# State-dependent binding of cholesterol and an anionic lipid to the muscle-type *Torpedo* nicotinic acetylcholine receptor
Anna Ananchenko[1], Rui Yan Gao[1], François Dehez [2]✉ & John E. Baenziger [1]✉

The ability of the *Torpedo* nicotinic acetylcholine receptor (nAChR) to undergo agonist-induced conformational transitions requires the presence of cholesterol and/or anionic lipids. Here we use recently solved structures along with multiscale molecular dynamics simulations to examine lipid binding to the nAChR in bilayers that have defined effects on nAChR function. We examine how phosphatidic acid and cholesterol, lipids that support conformational transitions, individually compete for binding with phosphatidylcholine, a lipid that does not. We also examine how the two lipids work synergistically to stabilize an agonist-responsive nAChR. We identify rapidly exchanging lipid binding sites, including both phospholipid sites with a high affinity for phosphatidic acid and promiscuous cholesterol binding sites in the grooves between adjacent transmembrane α-helices. A high affinity cholesterol site is confirmed in the inner leaflet framed by a key tryptophan residue on the MX α-helix. Our data provide insight into the dynamic nature of lipid-nAChR interactions and set the stage for a detailed understanding of the mechanisms by which lipids facilitate nAChR function at the neuromuscular junction.

The lipid composition of neuronal membranes is essential to healthy brain function, with changes in the levels of lipids correlating with aging and neurodegenerative disease[1,2]. In this broad medical context, it is intriguing that the activities of many proteins implicated in neuronal communication, including post-synaptic pentameric ligand-gated ion channels (pLGICs), are modulated by lipids[3–10]. Decades of biophysical and biochemical studies have established that the activity of the prototypic pLGIC, the muscle-type nicotinic acetylcholine receptor (nAChR) from *Torpedo*, is sensitive to a variety of lipids, including neutral lipids, such as cholesterol (Chol), and anionic lipids, such as phosphatidic acid (PA)[11–13]. Although the nAChR exhibits only weak structural specificity for both lipid types[14,15], PA is particularly effective at stabilizing an agonist-responsive conformation[16–18]. Also, a 3:1:1 molar ratio of phosphatidylcholine (PC), PA, and Chol is as effective as native membranes in stabilizing an agonist-responsive nAChR[13,19]. Lipids influence the magnitude of the agonist-induced response via a conformational selection mechanism whereby they stabilize different proportions of activatable resting versus non-activatable desensitized or uncoupled states[14,19,20]. Lipids also influence the kinetics of conformational transitions[21]. It remains to be determined whether lipids select conformations by binding to allosteric sites, altering bulk membrane-protein interactions, or a combination of both.

Recent cryo-electron microscopy (cryo-EM) structures determined in apo, agonist-bound, and toxin/inhibitor-bound states reveal densities in both the outer extracellular and inner cytoplasmic leaflets at the periphery of the TMD that have been attributed to bound lipids[22–27] (Fig. 1, see also ref. [28]). A conserved inner leaflet phospholipid site, modeled with bound PC, is located at each subunit-subunit interface with the headgroup phosphate framed by an arginine and histidine on the M3 α-helix from the principal subunit (+M3) and a positive lysine or arginine on the M4 α-helix from the complementary subunit (-M4)[22]. Both the absence of residues coordinating the modeled choline headgroup and the variability in head group poses from one subunit to another suggests that these sites may accommodate a variety of phospholipids, including PA. Diffuse density was also modeled as bound Chol, including sites deemed both high and low affinity, with the former occurring in the inner leaflet of the bilayer[24]. Densities attributed to Chol have also been observed in similar regions in cryo-EM images of the nAChR in native membranes[29,30]. Notably, some of the modeled phospholipid and Chol binding sites overlap while others do not. The structures raise the possibility that neutral and anionic lipids bind to multiple overlapping allosteric sites but with different affinities and thus potentially different efficacies for stabilizing activatable versus non-activatable conformations.

[1]Department of Biochemistry, Microbiology and Immunology, University of Ottawa, Ottawa, ON, Canada. [2]CNRS, LPCT, Université de Lorraine, F-54000 Nancy, France. ✉e-mail: Francois.Dehez@univ-lorraine.fr; John.Baenziger@uottawa.ca

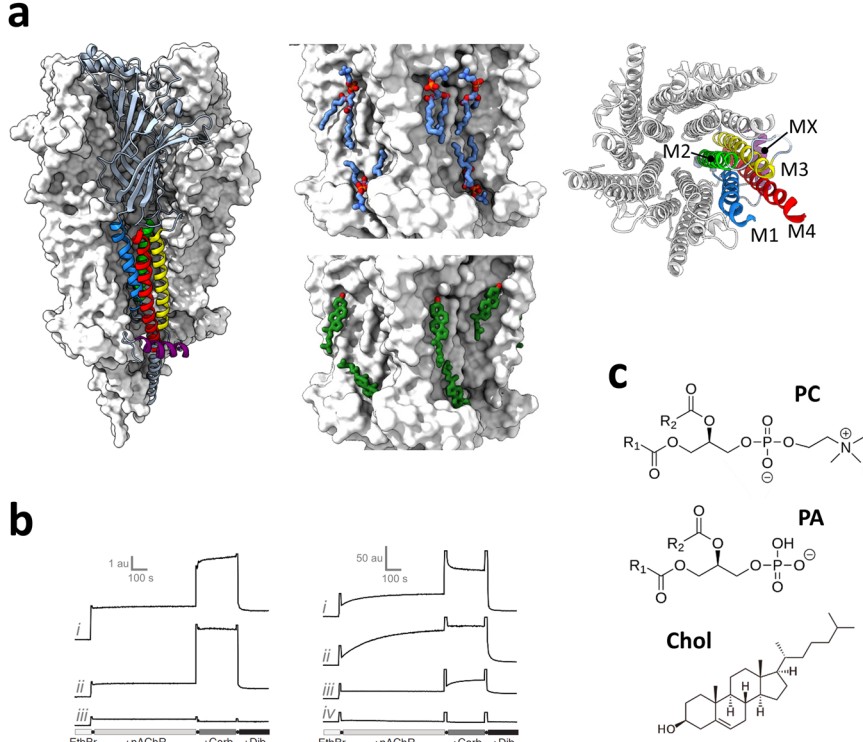

**Fig. 1 | Cryo-EM structures with bound lipids and functional lipid sensitivity of the nAChR. a** Structure of the nAChR with bound lipids. Left: side view of the nicotine-bound nAChR (PDB: 7QL5) represented as a white surface, with the α_γ subunit highlighted as cartoon colored by α-helices: M1, blue; M2, green; M3, yellow; M4, red; MX, purple. Middle top: Side view of the nicotine-bound nAChR TMD with bound PC shown as blue sticks. Middle bottom: Side view of the apo nAChR (PDB:7SMQ) with bound Chol shown as green sticks. Right: Top-down view of the TMD. **b** Kinetic binding of ethidium bromide (EthBr) assess the functional state of reconstituted nAChR membranes. EthBr binds to the TMD pore with an ~1000-fold higher affinity to *open* and *desensitized* rather than *resting* or *uncoupled* states, with its fluorescence increasing dramatically upon nAChR-

binding[81]. High affinity EthBr binding prior to agonist (carbamylcholine, Carb) addition likely reflects binding to pre-existing *desensitized* nAChRs. The addition of Carb transitions *resting* nAChRs to *open* and then *desensitized* states. The local anesthetic, dibucaine (Dib), competitively displaces EthBr binding. Left: the nAChR in *i*. native, *ii*. 3:1:1 PC:PA:Chol, and *iii*. PC-only membranes. Right: the nAChR in *i*. 3:2 PC:PA, *ii*. 3:2 PC:phosphatidylserine (PS), *iii*. 3:2 PC:Chol, and *iv*. PC-only membranes. Mixtures of PC:PA are more effective than mixtures of PC and other anionic lipids, such as PS, in stabilizing an agonist-responsive nAChR. Note that the left and right panels were acquired with different excitation/emission slit widths leading to different arbitrary fluorescence units (au). Data modified from refs. [19,20]. **c** Chemical structures of nAChR-interacting lipids.

Molecular dynamics (MD) simulations have been used to probe the dynamic interactions between lipids and the *Torpedo* nAChR using models ranging from individual transmembrane domain (TMD) α-helices imbedded in simple PC bilayers to the pioneering full-length 4 Å resolution *Torpedo* structure (PDB code, 2BG9) imbedded in complex neuronal-like membranes[31–35]. Although these models provided preliminary insight into the nature of lipid-nAChR interactions, the 2BG9 structure exhibits an error in the register of the amino acid sequence within the TMD and lacks the short MX α-helix in each subunit[23], the latter frames lipid binding to the above noted inner leaflet sites (Fig. 1). These and other errors alter how the modeled TMD interacts with its lipid environment thus limiting the conclusions derived from the simulation data.

With recent advancements in both computational methods and power, MD simulations are now poised to explore the dynamic nature of lipid-protein interactions, even with relatively large proteins such as the nAChR[36–38]. A coarse-grained MD (CG-MD) approach simplifies the system to reduce computational costs thus allowing simulation timescales long enough to sample in an unbiased manner both lipid association and dissociation events[39]. Individual frames from CG-MD simulations can be back-mapped to all-atom structures for simulations that shed light on the nature of the lipid-protein associations. MD simulations have been used to study lipid binding to a variety of pLGICs[35], with multi-timescale simulation approaches recently applied to the neuronal α7 nAChR[40], glycine receptor[41], acid-sensing ion channels[42], and the voltage-gated potassium channel, Kv1.2[43].

Here, we use a multiscale MD simulation approach to characterize dynamic lipid binding to both apo and nicotine-bound structures (2.9 and 2.5 Å resolution, respectively) of the *Torpedo* nAChR. We focus on simple bilayers, containing PC, PA and Chol, whose effects on nAChR function have been characterized in vitro, with the goal of elucidating how PA and Chol, two lipids that support conformational transitions, both individually and synergistically compete for binding to the nAChR with PC, a lipid that does not support conformational transitions. Our simulations reveal state-dependent interactions at subunit specific sites thus providing a framework for understanding the mechanisms by which lipids influence nAChR function.

## Results

As a first step towards understanding the mechanisms by which neutral and anionic lipids modulate *Torpedo* nAChR function, we performed $3 \times 30$ µs long CG-MD simulations with both apo and nicotine-bound nAChR structures imbedded in pure PC, 3:2 PC:PA, 3:2 PC:Chol, and 3:1:1 PC:PA:Chol membranes (all molar ratios). These lipid mixtures were chosen for three reasons. First, the agonist-induced response of the nAChR in each of the membranes has been characterized in vitro (Fig. 1). The nAChR in PC-only membranes is stabilized in an uncoupled conformation that normally does not undergo agonist-induced conformational transitions[14,19]. The 3:2 molar ratios of PC:PA and PC:Chol are the optimal ratios of both PA and Chol in a PC membrane for stabilizing an agonist-responsive nAChR, although mixtures of PC and PA are more effective in

this regard (Fig. 1)[14]. The 3:1:1 PC:PA:Chol membrane is as effective as native membranes at stabilizing an agonist-responsive nAChR and thus serves as a surrogate for a native membrane environment (Fig. 1b). Second, simulations of the nAChR in these four simple membranes allow us to directly assess how PA and Chol, lipids that in reconstituted membranes are particularly effective at supporting an agonist-induced response, both individually and synergistically compete for binding to the nAChR with PC, a lipid that does not support an agonist-induced response. Finally, the relatively high levels of PA and Chol enhance the sampling of lipid binding to provide unprecedented insight into the dynamic nature of PA and Chol interactions with the nAChR. We first describe the dynamic patterns of lipid binding to the apo nAChR followed by a discussion of state dependent lipid-nAChR interactions. We then explore the nature of PC, PA and Chol interactions using atomistic simulations performed on select frames back mapped from the CG-MD simulations.

## Lipid organization at the nAChR-lipid interface

2-D headgroup density projections (phosphate for PC and PA, hydroxyl for Chol) averaged over the combined set of three repeat trajectories show that all three lipids localize preferentially around the nAChR with concentric rings of diminishing density radiating out from the TMD (Figs. 2, 3, and S1) suggesting the nAChR influences lipid organization beyond a single shell of "boundary" lipids[44]. Localized regions of high headgroup density are observed in contact with the TMD. For the two phospholipids, side views show that these headgroup densities delineate the bilayer-water interface of both the outer and inner leaflets, although in a few select regions the head groups concentrate slightly higher to interact transiently with the Cys (β6-β7) loop of the extracellular domain or the β10-M1 linker. For Chol, additional head group density is located near the middle of the bilayer suggesting that the hydroxyl either bobs or dives into the membrane to interact with the TMD. The same global pattern of lipid organization around the nAChR, including both the concentric rings of lipid density radiating out from the nAChR and the local regions of high headgroup density immediately adjacent to the TMD is observed for both the apo and nicotine-bound structures.

## PC exchanges rapidly between nAChR-bound sites and the bulk membrane

In pure PC membranes, the PC headgroup in the outer leaflet localizes to numerous sites on each of the five roughly concave surfaces located between pairs of M4 α-helices from adjacent subunits, with lower lipid headgroup density surrounding the periphery of each of the M4 α-helices (Figs. 2, 3). Both the number of localized regions and their exact positions differ from one concave surface to another. Although these high-density regions reflect relatively high occupancy in that the phosphate is located at each site in a large percentage of simulation frames, the durations of binding are relatively short (Figs. 4, S2–S4) suggesting that the bound PC is in rapid exchange with PC in the bulk membrane.

   In the inner leaflet, PC exhibits longer duration binding to five inner leaflet pockets framed by the +M3 and +MX α-helices from the principal subunit and the -M1, -MX and -M4 α-helices from the complementary subunit. PC binds deeper (i.e., closer to the ion pore) to the α$_γ$-γ, α$_δ$-δ, and δ-β inner leaflet pockets with its phosphate coordinated by an arginine on +M3 (e.g., δArg315), a histidine located between +M3 and +MX (e.g., δHis320) and a positively charged lysine or arginine on -M4 (e.g., βArg437) (Fig. 4). PC binds more superficially to the β-α$_γ$ and γ-α$_δ$ inner leaflet sites with its phosphate positioned on the lipid-facing surface of +M3, with deeper access to the pockets sterically restricted by the C-terminus of -MX (Fig. 4b). Binding deeper to the site may also be less favorable because the positively charged lysine/arginine residue on -M4 is replaced by His (αHis408). In one prominent pose at the β-α$_γ$ and γ-α$_δ$ inner leaflet sites, the phosphate interacts directly with the +M3 arginine side chain and the choline projects downwards toward the groove between the adjacent +MX and -MX α-helices.

   The observed pattern of PC binding correlates with the pattern of phospholipid binding observed in cryo-EM structures[22,24]. As in the CG-MD

simulations, the structures also detect phospholipids (modeled as PC) bound to variable sites in the outer leaflet along with phospholipids bound to the same five pockets at subunit interfaces in the inner leaflet. The structures even capture PC binding deeper within the pocket at the α$_γ$-γ, α$_δ$-δ and δ-β interfaces. Both the MD simulations and the cryo-EM structures present a unified picture of phospholipid binding, albeit with the simulations showing that most of the bound PC exchanges rapidly with PC in the bulk membrane. A diagram depicting how the lipid binding poses of PC, PA and Chol detected in our simulations compare with the binding poses observed in cryo-EM structures is presented in Fig. S5.

## PA exhibits long-duration binding to subunit-selective inner leaflet pockets

The patterns of both PC and PA binding in 3:2 PC:PA membranes are similar to that of PC binding in pure PC membranes suggesting that the two phospholipids compete for overlapping sites (Figs. 2, 3 and S1). In the outer leaflet, both bind to numerous sites along each concave surface between pairs of M4 α-helices from adjacent subunits (Fig. S1) with the locations of these sites differing from one interface to another. The headgroup densities corresponding to PA are generally more intense than those of PC, particularly along the α$_γ$-γ and the α$_δ$-δ interfaces. PA binding to β10-M1 is more prominent in the δ subunit where two contiguous positively charged residues, δArg222 and δLys223, project towards the lipid bilayer (Fig. S6). The durations of these interactions for both PC and PA, however, are again relatively short.

   In the inner leaflet, PC and PA compete for the same five inner leaflet pockets but with different affinities (Fig. 3). PA selectively displaces PC from the pockets at the α$_γ$-γ, α$_δ$-δ, and δ-β interfaces. In fact, two PA molecules are observed binding to the α$_γ$-γ pocket in approximately 25% of the simulation frames. The enhanced occupancy of PA at these three sites is due primarily to longer binding durations (Figs. 4, S2, S6). For example, PA undergoes many interactions with the +M3 arginine that last between 5 and 15 μs, the latter corresponding to half of a 30 μs simulation trajectory. In contrast, PC maintains preferential binding to the inner leaflet pockets at the γ-α$_δ$ and β-α$_γ$ interfaces. As in the pure PC membranes, the bound PC still interacts extensively with residues in the loop between +M3 and +MX.

   PA exists in both monoanionic and dianionic forms, with the equilibrium between the two in a PC:PA 3:2 membrane exhibiting a pK$_a$ of 6.5[17]. While there are contrasting reports as to whether the monoanionic or dianionic forms of PA are responsible for its unique effects on channel function, we noted that the MARTINI v2.2 lipidome assigns PA a charge of −2. Given the cationic nature of the inner leaflet pockets, we repeated the CG-MD simulations using monoanionic PA. 2D headgroup density plots show that the monoanionic form of PA competes effectively with PC and thus binds deeply to the inner leaflet binding sites at the α$_γ$-γ, α$_δ$-δ, and δ-β interfaces, although binding is slightly more diffuse. As with dianionic PA, the enhanced affinity of PA for these three sites is due to its relatively long interaction durations (Fig. S7). Monoanionic PA competes more effectively with PC for binding to the γ-α$_δ$ and β-α$_γ$ inner leaflet pockets, although its ability to do so was not observed in all simulation repeats. The single cationic charge (γArg310 and βArg307) at the γ-α$_δ$ and β-α$_γ$ inner leaflet pockets may "solvate" monoanionic PA better than dianionic PA.

## Chol exhibits promiscuous long duration binding

The binding of PC in the 3:2 PC:Chol membranes is essentially identical to that observed in the absence of Chol. Chol has little effect on PC binding because the headgroup of the bound Chol is typically positioned lower and thus closer to the middle of the bilayer with the sterol ring penetrating the grooves between adjacent lipid-exposed α-helices, while the PC headgroup typically binds higher and closer to the aqueous interface in poses restricted to the lipid-exposed surfaces of the TMD α-helices. These poses are highlighted in the atomistic simulations discussed below (see Fig. S9). The flexible acyl chains of PC adapt around the bound sterol ring so that the two lipids collectively form a highly complementary interface with the TMD.

   In contrast, Chol exhibits a pattern of binding to the nAChR in 3:2 PC:Chol membranes that is strikingly distinct from the binding pattern of

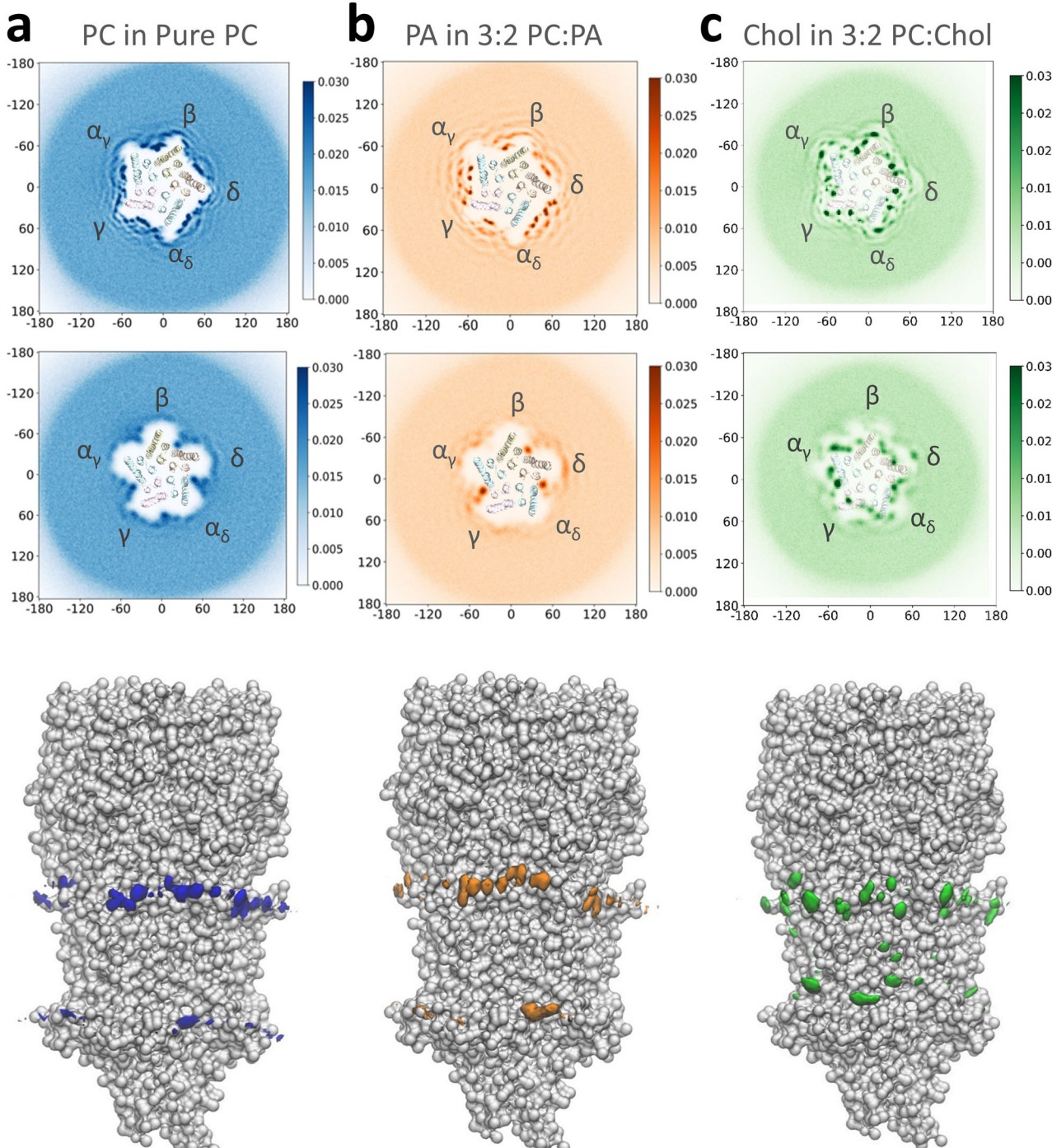

**Fig. 2 | PC, PA and Chol localization in simulations of the apo nAChR in membranes composed of PC, 3:2 PC:PA, and 3:2 PC:Chol.** The top two rows correspond to top-down 2D lipid headgroup density plots for (**a**) the PO$_4$ bead of PC, (**b**) the PO$_4$ bead of PA, and (**c**) the OH bead of Chol in each case averaged over 3 × 30 µs CG-MD simulations for the outer (top row) and inner (middle row) bilayer leaflets. The bottom row shows side views of the headgroup densities adjacent to the apo nAChR at the α$_\gamma$-γ interface, with the coarse-grained nAChR shown as silver van der Waals beads.

PC (Figs. 3, S1, S8). In the extracellular leaflet, Chol binding is centered around four main sites along each of the five concave surfaces between the M4 α-helices from adjacent subunits (Figs. 2, S1). Three of these sites are in contact with the principal face, with one of these located on the +M4 surface and the other two in the grooves between the +M4/+M3 and the +M3/-M1 α-helices. These three distinct poses are most clearly seen at the β-α$_\gamma$ and γ-α$_\delta$ interfaces. The fourth site is located on the complementary face in the -M3/-M4 groove, although binding to this site is variable leading to blurring of the headgroup density in the 2D density plots. Notably, the interactions

between non-polar residues and the sterol ring are typically longer than those involving the polar hydroxyl (Fig. 5). Together, these interactions lead to substantially longer Chol binding durations to the +M4/+M3 and +M3/-M1 grooves than to either the +M4 or the -M1/-M4 groove. Chol binding to the +M4/+M3 and +M3/-M1 grooves is also longer in duration than the binding of either PC or PA to their outer leaflet sites.

Chol binds to the same inner leaflet pocket that binds phospholipids, but with its hydroxyl located closer to the middle of the bilayer and deeper towards the ion pore (Fig. S5). Chol also binds on the principal face of the

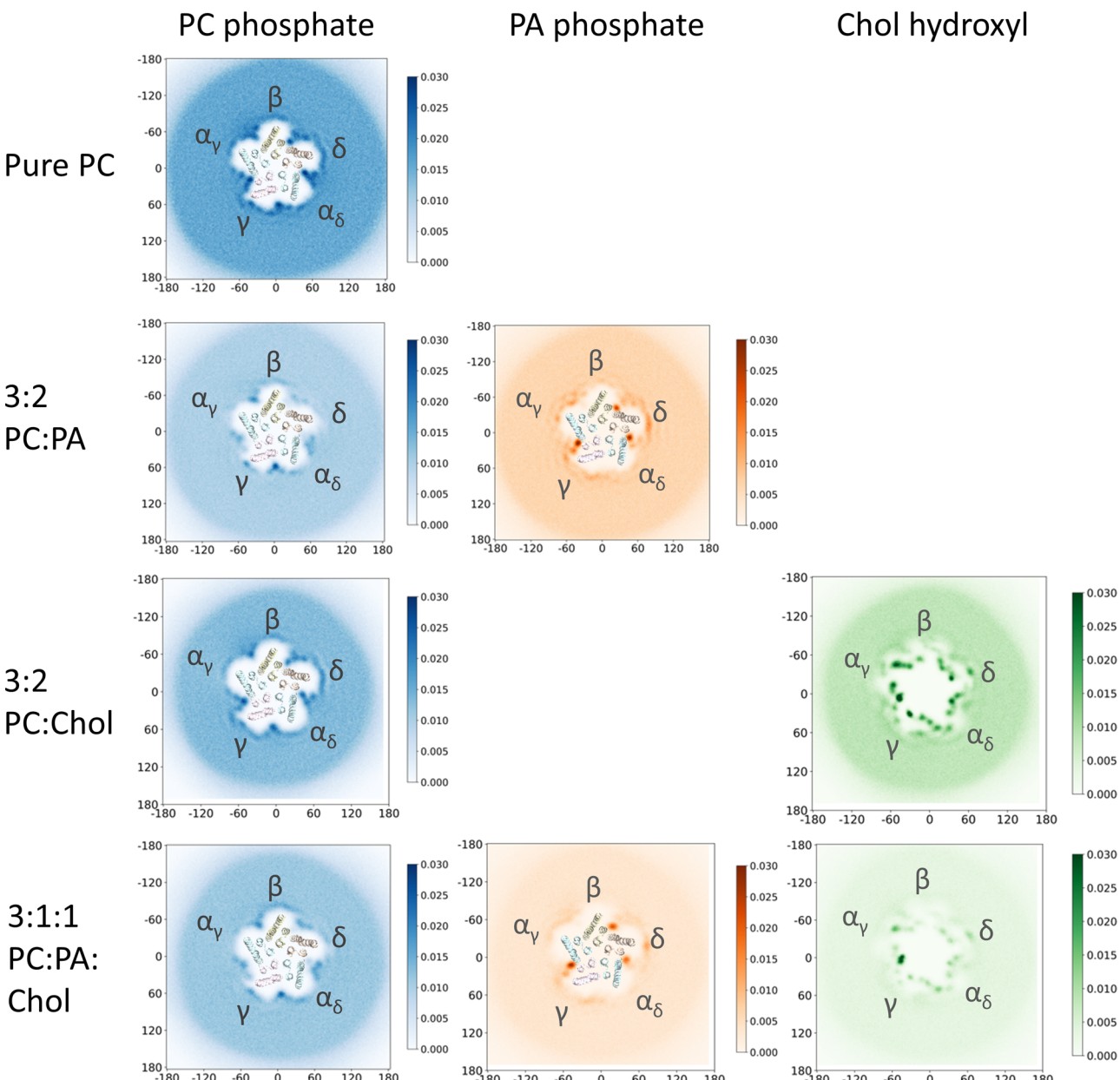

**Fig. 3 | Top-down 2D headgroup density plots for PC, PA and Chol in the inner leaflet from simulations of the apo nAChR.** Each density plot represents the lipid headgroup densities averaged over $3 \times 30\,\mu s$ CG-MD trajectories.

inner leaflet pocket with its sterol positioned in the +M4/+M3 groove and its hydroxyl buried in a pocket framed by the +M4, +M3, and +MX where it can interact with the same conserved +M3 arginine that plays a critical role in phospholipid binding. The framing of this pocket at the $\alpha_\gamma$-$\gamma$, $\alpha_\delta$-$\delta$, and $\beta$-$\alpha_\gamma$ interfaces is facilitated by a bulky tryptophan (e.g., $\alpha$Trp311) on +MX, with Chol binding at these three sites exhibiting higher occupancy and longer interaction durations than binding to the same sites at the $\gamma$-$\alpha_\delta$, and $\delta$-$\beta$ interfaces. High affinity Chol binding is observed at these $\alpha_\gamma$-$\gamma$, $\alpha_\delta$-$\delta$, and $\beta$-$\alpha_\gamma$ inner leaflet pockets in cryo-EM structures (Fig. S3).

In addition to binding within the outer and inner leaflets, the Chol hydroxyl often bobs up/down or dives into the membrane to interact with the TMD, the latter orienting the sterol ring roughly parallel to the bilayer surface (Fig. S9). Supplemental movie 1 captures a single Chol in the outer leaflet that binds to the TMD, bobs down into the inner leaflet, flips its orientation, and then dives up so that its hydroxyl interacts with $\gamma$Ser235 for the remainder of the simulation trajectory. Surprisingly, the Chol hydroxyl interacts frequently with numerous polar and hydrophobic residues within the TMD suggesting that bobbing and diving events for Chol are common. The extensive sampling of the lipid exposed TMD surface facilitates movements of Chol from one leaflet of the bilayer to the other, although it does not change the distribution of Chol in either leaflet (Fig. S10).

**Synergistic effects of PA and Chol**
Although most of the binding sites for PC, PA, and Chol are maintained in the 3:1:1 PC:PA:Chol membrane, many exhibit reduced occupancies. In the outer leaflet, both PC and PA still bind to multiple sites along each concave surface between the M4 $\alpha$-helices from adjacent subunits, with PA binding slightly more prevalent than PC (Fig. S1). The interactions of PC and PA in the outer leaflet are still of relatively short duration and likely represent non-specific binding. Many of the localized regions of Chol binding to the outer leaflet also exhibit reduced occupancy. For example, although three distinct Chol binding poses are observed along the principal face of both $\beta$-$\alpha_\gamma$ and $\gamma$-$\alpha_\delta$ in the 3:2 PC:Chol membranes, these poses are only observed at the $\gamma$-$\alpha_\delta$ interface in the 3:1:1 PC:PA:Chol membrane.

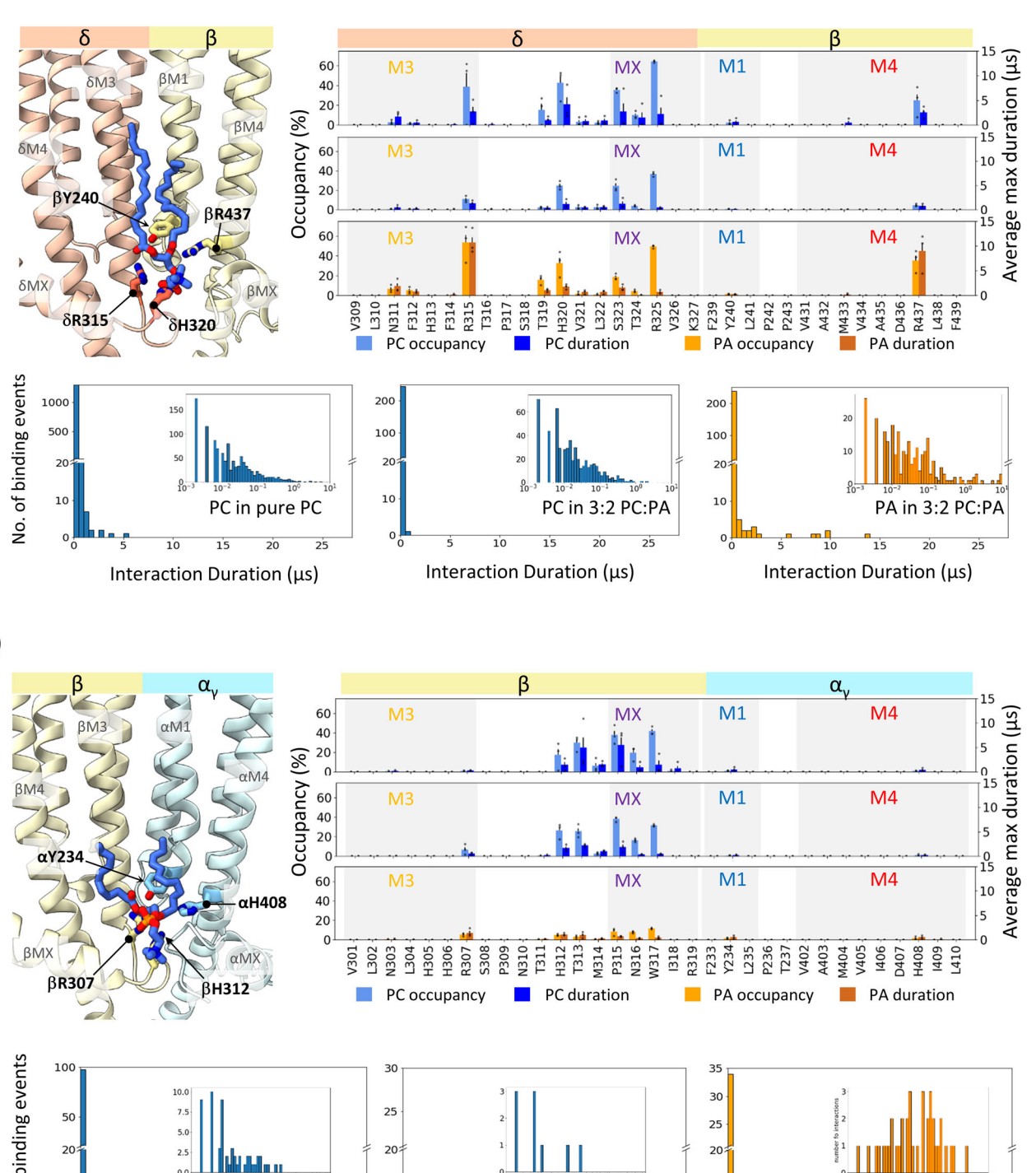

**Fig. 4 | Phospholipid interactions at "high" and "low" affinity PA inner leaflet binding sites of the apo nAChR. a** Top left panel shows the bound PC modeled at the δ-β inner leaflet site of the apo nAChR (PDB: 7QKO) with the subunits shown as cartoons and both lipid and coordinating residues shown as sticks. Top right dual-axis plots showing the % occupancy (% of frames with bound lipid) and average maximum durations (average of the longest interaction duration from each of the three simulation trajectories) from CG-MD simulations of lipid binding to the δ-β inner leaflet site. Error bars represent standard deviations. Overlaid points represent occupancy or maximum duration values from each repeat. Interactions were determined using the dual cutoff method (see Methods). The top plot summarizes PC binding in pure PC membranes while the middle and bottom plots summarize PC and PA binding, respectively, in 3:2 PC:PA membranes. The bottom three panels in (**a**) are histograms depicting the interaction durations of each lipid with δR315 using both linear and log (inset) time scales. **b** Same data as in (**a**) but for lipid binding to the low PA affinity β-αγ inner leaflet site.

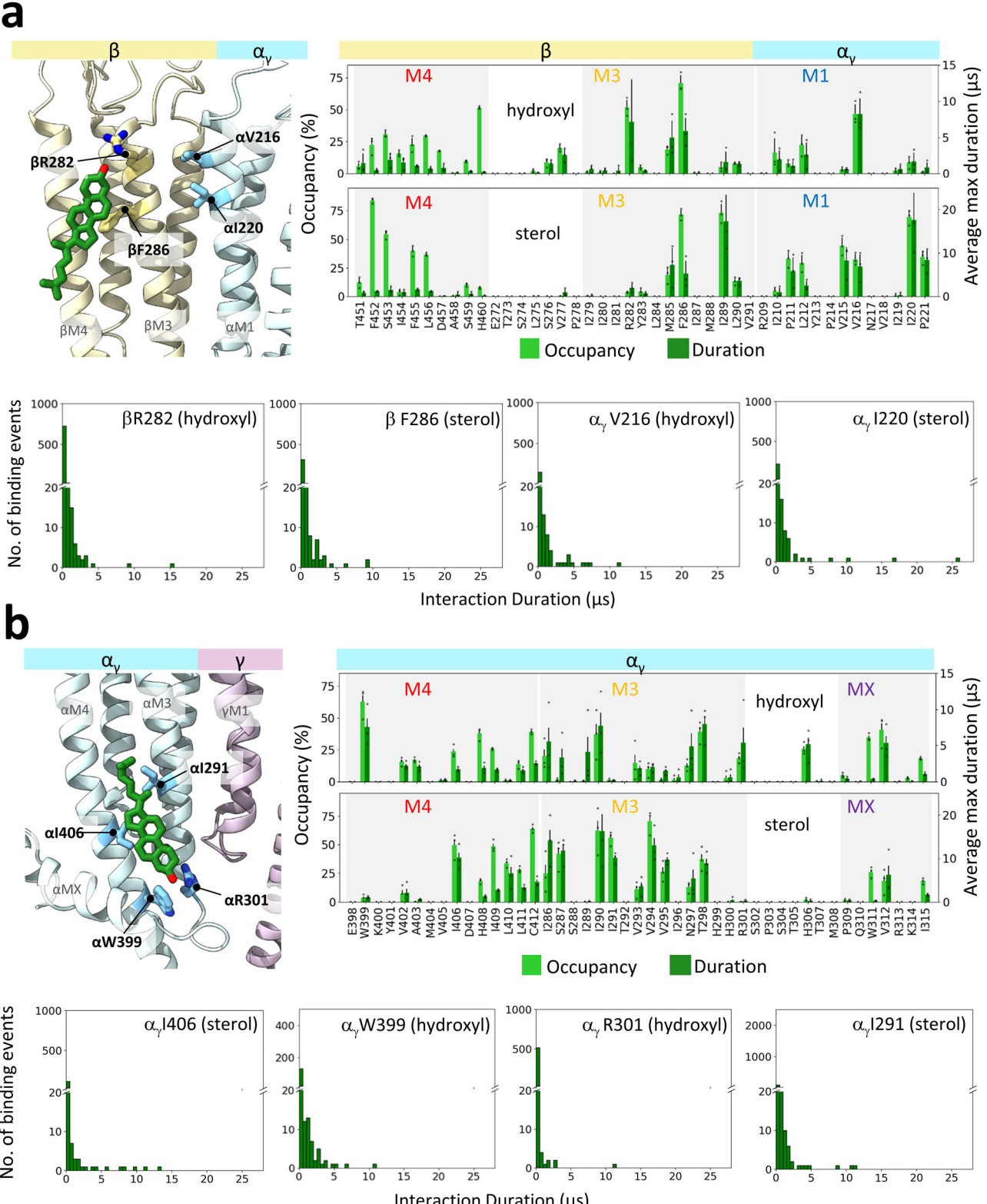

**Fig. 5 | Chol interactions at upper and lower leaflet sites in PC:Chol 3:2 membranes. a** Top left panel shows Chol bound at the β-α_γ upper leaflet site in the structure of the apo + Chol nAChR (PDB: 7SMQ). The subunits are shown as cartoon, while the lipid and Chol-interacting side chains are shown as sticks. Top right dual-axis plots showing % occupancy and average maximum durations (see Fig. 4) for the Chol hydroxyl and sterol ring interacting with residues at the same β-α_γ interface. Error bars represent standard deviations. Overlaid points represent occupancy or maximum duration values from each repeat. The bottom four panels in (**a**) show interaction duration histograms for the Chol hydroxyl with βR282 and αV216 and for the Chol sterol ring with βF286 and αI2220. **b** Same data as in (**a**) but for Chol interacting with the inner leaflet site at the α_γ-γ interface. The interaction duration histograms are for the Chol hydroxyl interacting with α_γW399 and α_γR301, and the Chol sterol interacting with α_γI406 and α_γI291.

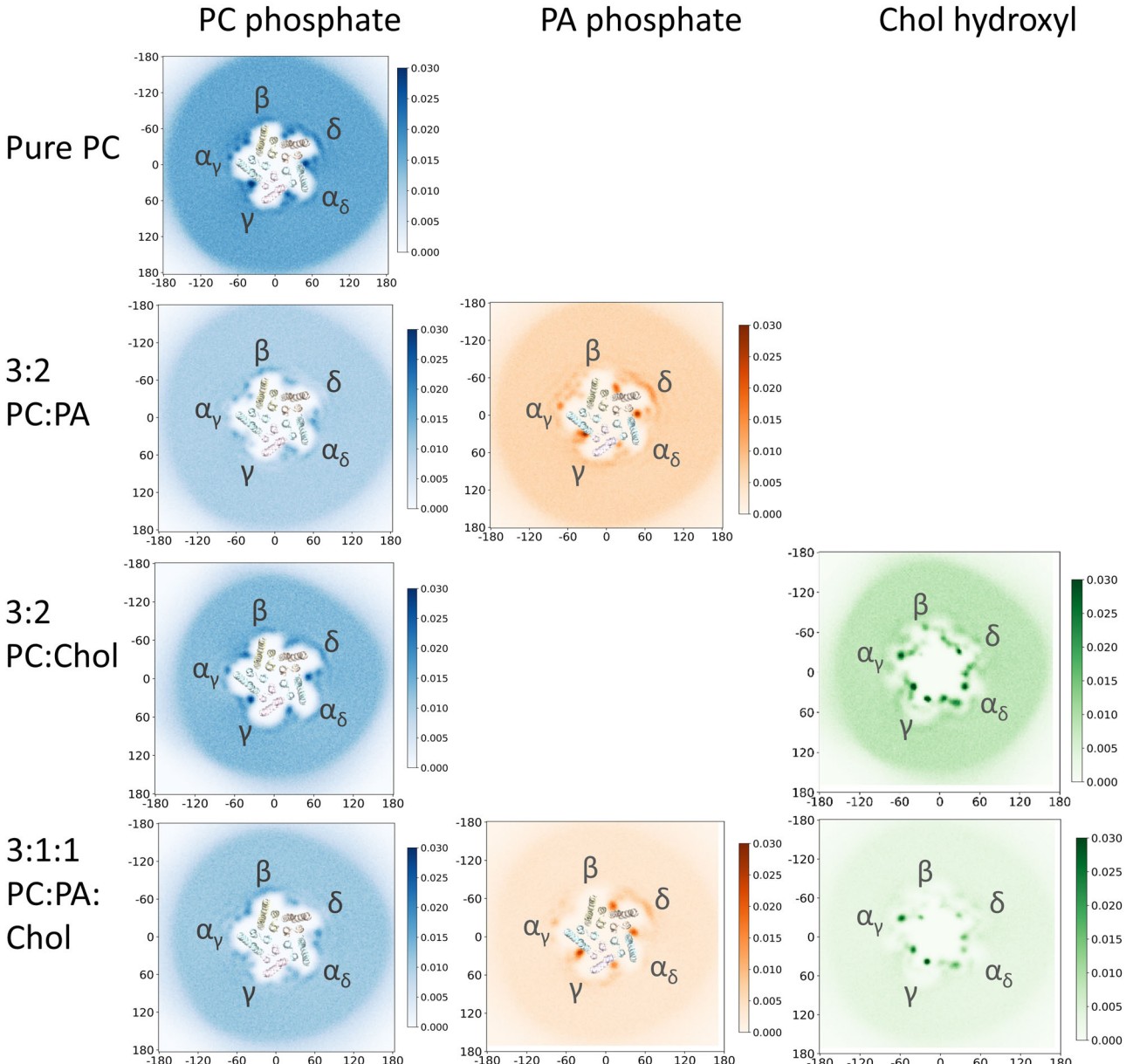

**Fig. 6 | Top-down 2D headgroup density plots for PC, PA and Chol in the inner leaflet from simulations of the nicotine-bound nAChR.** Each density plot represents the lipid headgroup densities averaged over 3 × 30 μs CG-MD trajectories.

Notably, PC or PA still bind preferentially to the inner leaflet pockets, with PA displacing PC at the α$_γ$-γ, α$_δ$-δ, and δ-β sites. Both the intensities of PA and PC binding to these sites (Fig. 3) and their interaction durations are similar to those observed in the 3:2 PC:PA membrane. In contrast, the majority of the Chol binding events in the inner leaflet scale roughly proportionally with the reduced levels of Chol, the one exception being Chol binding to the principal face at the α$_γ$-γ interface. The ability of Chol to maintain binding highlights a potential modulatory role for this site in Chol action at the nAChR.

**State dependent lipid interactions**

We repeated the CG-MD simulations with the nicotine-bound structure of the nAChR, which has a capped loop C around the agonist and an open Leu9'/Val13' gate. The patterns of lipid binding to the apo and nicotine-bound structures are similar (Fig. 6), with three notable exceptions:

First, although PC and PA exhibit a similar pattern of binding to the α$_γ$-γ, α$_δ$-δ, δ-β and β-α$_γ$ inner leaflet pockets as in the apo state, subtle agonist-induced movements of the α$_δ$ MX α-helix allow PC to penetrate

deeper into the γ-α$_δ$ inner leaflet pocket in both the pure PC and 3:2 PC:Chol membranes leading to slightly longer interaction durations. The subtle opening of the γ-α$_δ$ inner leaflet pocket also allows PA to penetrate more effectively and out compete PC for binding.

Second, nicotine binding subtly alters the pattern of Chol interactions with the TMD of each subunit. For example, Chol interactions in the inner leaflet at the α$_γ$-γ interface are longer in duration and are centered on a more condensed region in the apo versus the nicotine-bound state, while interactions at the same site at the α$_δ$-δ interface are similar in both states.

Third, a new site for Chol binding to the nicotine-bound state is observed deep within the TMD at the β-α$_γ$ interface. This site corresponds to Chol diving into the bilayer so that its hydroxyl penetrates an enlarged hydrophobic pocket framed by βAla292, βIle296 and α$_γ$Leu228 to interact with βLeu256 on the lipid-facing side of the pore-lining M2 α-helix (Fig. S11). Notably, the allosteric potentiator PNU binds to a similar site on the α7 nAChR, with binding possibly stabilizing the open state[45] MD simulations show that Chol binds to the same PNU site on the desensitized

**Fig. 7 | PA and Chol binding characterized by atomistic simulations. a** A time-course plot of the distances between one phosphate oxygen of PA and the nitrogen of both αR301 and δR355 over the course of one 250 ns atomistic simulation trajectory of the apo nAChR in a 3:2 PC:PA membrane. The starting configuration was back-mapped from a CG-MD simulation. "Swarm images" showing different binding poses over the course of one 250 ns atomistic simulation trajectory are presented for (**b**) PA binding to the α_δ-δ inner leaflet site in 3:2 PC:PA membranes, (**c**) Chol binding to the α_γ-γ inner leaflet site in 3:2 PC:Chol membranes, and (**d**) Chol binding to the β-α_γ outer leaflet site in 3:2 PC:Chol membranes. Each swarm image reflects the position of the lipid and coordinating residues at 8 ns intervals (200 simulation frame intervals) over the course of a 250 ns atomistic simulation. Images were aligned on the relevant lipid-interacting transmembrane helices.

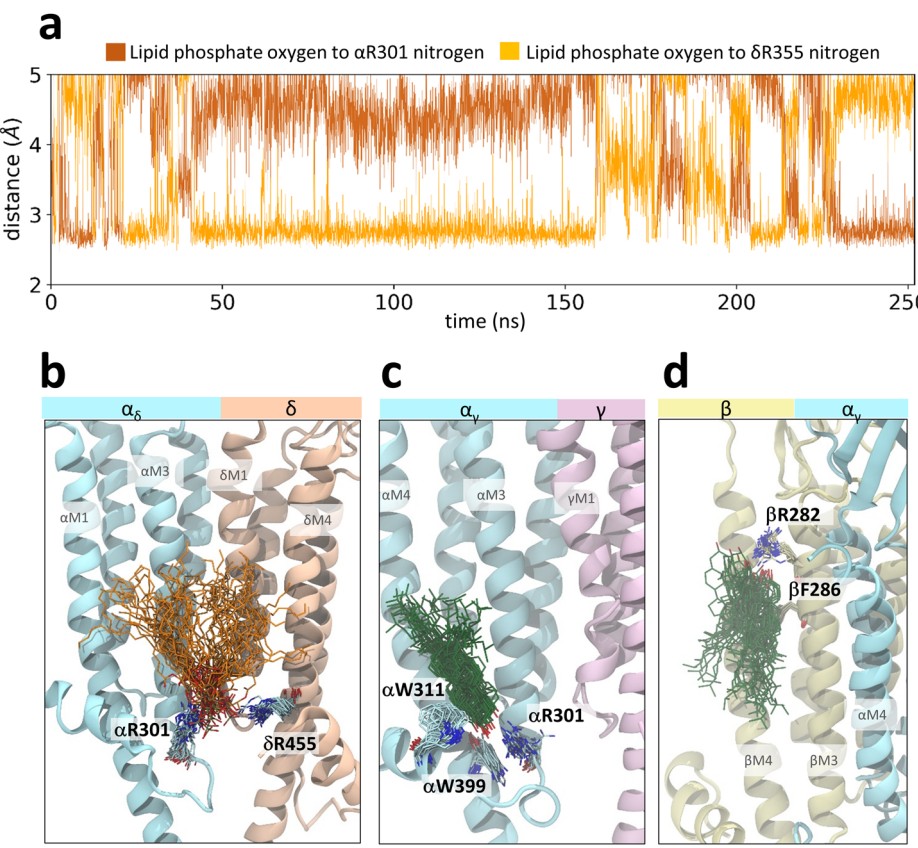

α7 nAChR[40]. The binding of Chol to this hydrophobic pocket on the *Torpedo* nAChR could help stabilize an agonist-responsive nAChR.

Finally, although previous structures have revealed a tilting of the M4 α-helices in the agonist-bound state[22,24], which could lead to enhanced lipid binding at the M4-M1/M3 interface, we did not detect major changes in either the lipid occupancy or interaction durations at this interface in the nicotine bound state.

## Atomistic simulations

To explore the nature of the physical interactions that drive lipid binding, we back mapped select frames from the apo and nicotine-bound CG-MD simulations and performed 2 × 250 ns long all atom simulations. We focus here on atomistic simulations of the nAChR in PC, 3:2 PC:PA and 3:2 PC:Chol membranes as these highlight the nature of the interactions between each lipid and the nAChR. For PC and 3:2 PC:PA, we back-mapped frames where PC or PA, respectively, is bound to each of the three α_γ-γ, α_δ-δ, and δ-β inner leaflet pockets and to at least one of the γ-α_δ or β-α_γ inner leaflet pockets. The all-atomistic simulations were performed with PA in the monoanionic state. For the 3:2 PC:Chol membrane, we back mapped frames that had the largest number of residues with both Chol occupancies above 40% and average longest Chol interaction durations >3 µs.

All but one of the six PC molecules and all six PA molecules remain bound to the α_γ-γ, α_δ-δ, and δ-β inner leaflet pockets (two repeats of three sites) over the course of the 250 ns simulation trajectories. For both lipids, one of the headgroup phosphate oxygen atoms remains within 3 Å (a distance indicative of a salt bridge) of one of the nitrogen atoms on either the +M3 arginine or the -M4 arginine/lysine essentially throughout the simulation trajectory (Fig. 7). A similar PA interaction has been proposed with the α2β4 nAChR[46]. At the same time, the phosphate and glycerol backbone oxygens form transient interactions with polar residues, including αHis306 and αAsn297, respectively, while the acyl chains interact with previously identified lipid-coordinating residues on -M1, such as γPhe241

and βTyr240. The choline headgroup of PC is relatively dynamic and does not form long duration interactions with any side chain. The all-atom simulations suggest that an oscillating salt bridge between the lipid phosphate and either the +M3 arginine or the -M4 lysine/arginine is the main stabilizing interaction at each of these three sites.

Although PC or PA bound to the γ-α_δ and β-α_γ interfaces in the apo state still form interactions of less than 3 Å distance with the + M3 arginine, these are sporadic. The PC phosphate also binds transiently to γSer316/βThr313 with the choline headgroup simultaneously interacting with γGlu319 at the γ-α_δ interface – interactions at β-α_γ were not well sampled due to frame selection. Notably, the –MX helix at the γ-α_δ interface is shorter and the inner leaflet pocket more "open" in the nicotine-bound state thus allowing PA to form a salt bridge with the +M3 arginine that lasts roughly half of each simulation trajectory. This long duration interaction accounts for the ability of PA to outcompete PC for binding to this site in the agonist-bound state.

Chol binding appears to be governed by non-specific van der Waals and polar interactions involving the sterol ring and the hydroxyl, respectively. Chol binding to a β-α_γ outer leaflet site and the α_γ-γ inner leaflet pocket highlight the general nature of Chol-nAChR interactions (Fig. 7c, d). At the β-α_γ outer leaflet site, one trajectory captures Chol initially adopting a pose with its sterol ring oriented diagonally across the +M4/+M3 groove and its hydroxyl interacting with βArg282 (Supplementary Movie 2). Midway through the simulation, however, the sterol ring inserts into the +M4/+M3 groove, with this parallel orientation over the course of the simulation eventually breaking the interaction between the hydroxyl and βArg282. Such competing sterol ring/hydroxyl interactions were observed frequently in the atomistic simulations. In contrast, the binding pose of Chol at the α_γ-γ inner leaflet is relatively stable over the entire simulation trajectory likely because the sterol ring is tightly sandwiched between the +M4/+M3 groove and +MX. In this pose, the Chol hydroxyl simultaneously forms transient interactions with several residues including αArg301, αHis306 and αTrp399.

## Discussion

Here, we have taken a unique multiscale MD simulation approach to study lipid binding to the nAChR in that we have focused on simple, defined membranes containing levels of PC, PA and/or Chol that are known to stabilize the nAChR in reconstituted membranes in different conformational states. These simulations allow us to correlate lipid binding patterns directly with documented effects of lipids on the function of the nAChR in reconstituted liposomes. Specifically, we set out to understand how lipids, PA and Chol, that support an agonist-induced response compete for binding to the nAChR with a lipid, PC, that does not. The relatively high levels of PA and Chol also enhance the sampling of lipid binding. The simulations provide direct insight into the dynamic nature of lipid binding to both the apo and agonist-bound *Torpedo* nAChR that is complementary to the static snapshots of lipid binding detected in cryo-EM structures (Fig. S5). The resulting data not only set the stage for functional measurements, which should provide detailed insight into the mechanisms by which lipid binding influences nAChR function, but also reconcile observations from previous biophysical studies that are foundational to our understanding of lipid-nAChR interactions.

First, simulations detect sites of lipid binding on the surface of the TMD, although most of the bound lipids exchange rapidly with lipids in the bulk membrane. Notably, PA and Chol bind to sites with longer durations and thus higher occupancy than PC. Both observations are consistent with electron spin resonance experiments that detect rapidly exchanging nAChR-bound lipids in both native and reconstituted membranes, albeit with larger fractions of immobilized PA and Chol[44,47]. In addition, the electron spin resonance experiments estimate rates of exchange between the bulk and immobilized lipids between $10^5\,s^{-1}$ and $5 \times 10^7\,s^{-1}$ [44,47], which are consistent with the ns to µs lipid residence times detected in the MD simulations (Figs. 4 and 5). The MD simulations and electron spin resonance data thus present a unified snapshot of lipid-nAChR interactions.

Second, simulations address a long-standing controversy regarding the existence of "non-annular" Chol binding sites on the nAChR. These sites located deep within each subunit's α-helical TMD bundle were originally proposed based on the observation that the nAChR's intrinsic tryptophan fluorescence is quenched to a greater extent by brominated Chol than by brominated PC, suggesting that Chol accesses sites that are PC inaccessible[48]. Based on the early 4 Å resolution 2BG9 *Torpedo* structure, it was suggested that Chol fits into cavities between TMD α-helices to stabilize the TMD, although the detected cavities are *absent* in higher resolution nAChR structures (Fig. S12)[33]. While neither the cryo-EM structures nor our MD simulations support the existence of deeply buried non-annular Chol sites, the simulations show that the relatively small sterol ring of Chol penetrates the shallow grooves between adjacent lipid-exposed transmembrane α-helices while the larger phospholipids are restricted to the lipid-exposed α-helical surfaces (Fig. S9). The simulations also capture Chol bobbing and diving into the membrane to interact with the TMD near the middle of the bilayer. These unique, albeit bulk lipid exposed, Chol binding poses likely account for the enhanced fluorescence quenching by brominated Chol.

Third, the observed Chol binding patterns shed light on the general nature of Chol-protein interactions. It has been suggested that Chol binding to membrane proteins is localized by CRAC ((L/V)-X$_{1-5}$-(Y)-X$_{1-5}$-(K/R)) and CARC (K/R)-X(1,5)-Y-X(1,5)-[L/V] motifs[49,50]. The locations of these putative motifs on the nAChR, however, do not correlate with the prominent sites of Chol binding observed in our simulations (Fig. S13). In fact, although we observe that Chol binds promiscuously over the entire lipid exposed TMD surface, high occupancy/long duration interactions occur in grooves between adjacent TMD α-helices – i.e., the sites for Chol binding are framed by residues that are on adjacent α-helices and that are thus distant from each other in the nAChR sequence. Binding sites are typically formed by a patch of hydrophobic residues (contributions from residues located on adjacent α-helices), which interact with the sterol ring, and a polar side chain, which interacts with the hydroxyl. Long duration binding is typically driven by van der Waals interactions between the hydrophobic residues and

the sterol ring. Although both CRAC and CARC motifs fit the general pattern required for Chol binding, the simulations suggest that a *local* sequence specific Chol binding motif does not exist on the nAChR.

Fourth, the simulations and structural data reveal numerous binding sites for both phospholipids and Chol that likely facilitate roles for lipids in nAChR folding and/or function (Fig. S3), although the correlations between lipid binding and channel function are likely complex. The phospholipids, PC and PA, compete for multiple binding sites, with both lipids exhibiting high occupancy at five inner leaflet pockets framed by the +M3 and +MX α-helices from the principal subunit and the -M1, -MX and -M4 α-helices from the complementary subunit. High occupancy Chol binding is also observed in both the outer and inner leaflets, most notably on the principal face of the same inner leaflet pockets that bind phospholipids. Significantly, both PC and Chol binding is modeled at the same sites in cryo-EM structures[22,24]. The simulation and structural data thus provide cohesive evidence for high affinity phospholipid and Chol binding to the nAChR.

The inner leaflet phospholipid binding pockets may serve as allosteric sites that regulate folding and/or function. Both PC and PA bind to the $\alpha_\gamma$-γ, $\alpha_\delta$-δ, and δ-β inner leaflet pockets with their phosphates coordinated by a conserved arginine on +M3 and a positively charged lysine or arginine on -M4. In both cases, the bound lipid phosphate oscillates between forming a salt bridge with one or the other positively charged residue. Significantly, a steric clash between the large choline headgroup and the -MX α-helix renders these interactions with PC relatively unstable. In contrast, the small headgroup facilitates PA access to these three sites leading to substantially longer interactions indicative of high affinity binding, consistent with previous MD simulations showing anionic lipid binding in the same region of the 2BG9 *Torpedo* structure[35]. PA may be more effective than other phospholipids in stabilizing an agonist-responsive nAChR because its small headgroup facilitates access to these sites leading to high affinity binding, with such high affinity binding possibly required to stabilize an agonist-responsive resting conformation.

Alternatively, the preferential binding of PC to the β-$\alpha_\gamma$ and γ-$\alpha_\delta$ inner leaflet pockets may favor the non-responsive uncoupled nAChR. These sites lack a positively charged arginine/lysine residue on -M4 and exhibit a unique conformation of -MX that limits phospholipid binding deep in the pocket. Phospholipids bind to these two inner leaflet pockets with their phosphates typically positioned on the lipid exposed side of the +M3 α-helix. Structures of the uncoupled nAChR are required to fully understand whether preferential PC binding to these sites favors an uncoupled state.

The MD simulations suggest several potential mechanisms by which Chol could work synergistically with anionic lipids to stabilize agonist responsive nAChRs. Chol may influence function by binding close to the same high affinity $\alpha_\gamma$-γ, $\alpha_\delta$-δ, and δ-β inner leaflet pockets to facilitate PA binding. Chol may also bind to the principal face of the $\alpha_\gamma$-γ inner leaflet pocket where it exhibits high occupancy even in the presence of both PC and PA. The synergistic effects of Chol on nAChR function could also arise from Chol binding to other regions of the TMD surface. The simulations highlight a propensity for the sterol ring to bind to the shallow grooves between TMD α-helices. Such binding could easily influence packing of the TMD α-helices in a manner that favors one conformation over another or that influences the relative movements of TMD α-helices during conformational transitions. In addition, Chol frequently bobs and dives into the membrane to interact with residues on the TMD surface.

Although a complete map of the nAChR conformational landscape remains to be defined, we observe numerous conformational specific interactions that could further underlie the effects of PA and Chol on nAChR function. Agonist-binding leads to movement of the -MX α-helix that open slightly the γ-$\alpha_\delta$ interface allowing PC to penetrate deeper in the pocket leading, with PA competing with PC for binding in the nicotine bound state. Enhanced PA interactions at this site could promote channel function. Especially interesting are the instances where the hydroxyl of Chol dives into the membrane to insert deeply at the β-$\alpha_\gamma$ interface to contact residues on M2 in the nicotine bound state. The binding of Chol to a similar M2 site has been observed in a state-dependent fashion in CG-MD

simulations of the GlyR[41] and the α7 nAChR[40]. The mechanistic importance of these and other observed lipid binding sites must be verified by functional measurements.

Finally, our data suggest features that likely impact on our understanding of lipid interactions at other pLGICs, including prokaryotic pLGICs, such as GLIC and ELIC[51,52]. PIP$_2$ and PE/PG bind to similar inner leaflet pockets on the GABA$_A$R[53] and on ELIC[54], respectively. The binding of phospholipids with larger headgroups to these two pLGICs is likely facilitated by the absence of MX α-helices, which frame the inner leaflet sites in the nAChR. In fact, the MX α-helices undergo subtle conformational changes that may impact on state dependent lipid binding to the nAChR. It has been suggested that phospholipid binding to outer leaflet sites on ELIC plays a role in channel function[55]. While such binding is important in ELIC, our simulations suggest that phospholipid binding to outer leaflet sites on the nAChR are of short durations and thus likely reflect non-specific binding.

## Methods
### Molecular samples
We focused our MD simulations on recently published structures of the nAChR[22] solved in the apo (resting state, PDB: 7QKO) and nicotine-bound (desensitized-like with a capped loop C around the agonist and an open Leu9'/Val13' gate, see Discussion in Zarkadas et al.[21], PDB: 7QL5) states at 2.9 Å and 2.5 Å resolution, respectively. Glycosylation sites were removed from the structures, and nicotine was removed from the nicotine-bound structure. The insane.py script[56] was used to embed the CG models in a bilayer in a 180 × 180 × 200 Å³ cubic box, solvated in MARTINI CG water with 0.15 mM NaCl, with additional neutralizing Na+ ions added to systems containing phosphatidic acid. Insane.py randomly places lipids in the intra- and extracellular leaflets, thus three repeats of simulations were set up independently to randomize lipid placement. Each structure was embedded in membranes composed of PC, 3:2 mol:mol PC/PA, 3:2 mol:mol PC/Chol, and 3:1:1 mol:mol:mol PC:PA:Chol. Membrane compositions were selected based on known lipid ratios that stabilize resting-to-desensitized conformational transitions of the nAChR, as discussed in the Results and shown in Fig. 1[14,19]. Simulation systems are summarized in Tables S1 and S2.

### Fluorescence experiments
Ethidium bromide fluorescence experiments were performed on a Cary Eclipse fluorescence spectrophotometer (Varian Inc.)[19,20]. The fluorescence intensity at 590 nm was monitored as a function of time (2.0 s sampling time), while ethidium was excited at 500 nm. At the indicated times, ~250 nM (left panel) or ~50 nM (right panel) nAChR, 500 μM Carb, and 500 μM dibucaine were added to a 0.3 μM solution of ethidium bromide in *Torpedo* Ringer buffer (5 mM Tris, 250 mM NaCl, 5 mM KCl, 2 mM MgCl$_2$, and 3 mM CaCl$_2$, pH 7.0). The data in the left and right panels were acquired with ±5 nm and ±20 nm excitation/emission slits, respectively.

### Coarse-grained simulations
The PDB files, with ligands and bound lipids removed, were converted to coarse-grain (CG) representation using the martinize.py script and the MARTINI2.2 CG forcefield[39]. The 3D structure of the CG proteins was restrained using the ElNeDyn network[57] using the default force constant of 500 kJ mol-1 and default lower and upper cutoffs of 0.5 and 0.9 nm respectively. The GROMACS[58–60] 2021 simulation engine was used for CG simulations. Soft-core energy minimization was performed for 5000 steps, then double-precision minimization for an additional 5000 steps using the steepest descent algorithm, or until a maximum force of 1000 kJmol−1 nm −1 on any atom was reached. Systems were equilibrated using the NPT ensemble for a total of 4.75 ns with gradual lowering of protein position restraints from 1000 to 50 kJ. Temperature held at 310 K during equilibration steps using the v-rescale thermostat[61] and pressure was held at 1 bar with the Berendsen barostat[62]. Production runs were performed for 30 μs each with a timestep of 20 fs using an NPT ensemble with v-rescale set to 310 K and semi-isotropic Parrinello-Rahman pressure coupling[63] set to 1 bar. Three simulation repeats were performed for each system.

### Atomistic simulations
A single frame was selected from each CG simulation repeat for each system, where lipids of interest were bound to possible identified binding sites. The CG2AT script[64] was used to backmap the system to an atomistic representation, with the box size reduced to 130 × 130 × 190 Å³, resulting in a system size of approximately 300,000 atoms. The original cryo-EM structures were used for steered-MD alignment of the backmapped coarse grained structure as described in[64].

Proteins, lipids and ions were modeled using the CHARMM36 force field[65], and water was described by the TIP3P model[66]. Salt concentration was again brought to 0.15 M of NaCl, with some ions added to neutralize systems containing PA. The NAMD 3.11 simulation software[67] was used for atomistic simulations. The backmapped systems were minimized for 1000 steps. Systems were equilibrated for 5 ns with soft harmonic positional restrains on all heavy protein atoms, followed by 5 ns of equilibration with positional restraints on heavy backbone atoms only. Production runs were performed with no positional restrains for two repeats of 250 ns for each backmapped system. Hydrogen mass repartitioning[68] was used for all atomistic simulations, allowing for a timestep of 4 fs. A timestep of 8 and 4 fs was used for long- and short-range interactions respectively, using the r-RESPA multiple time-stepping algorithm[69]. Temperature was held constant at 300 K using the Langevin thermostat[70] and pressure was held at 1 atm using the Langevin piston method. The SETTLE algorithm[71] was used for water and the SHAKE/RATTLE algorithms[72,73] were used for constraining covalent bonds involving hydrogen atoms to their experimental lengths.

### Analysis
Lipid density was calculated using built-in functions within the package MDAnalysis[74,75]. Briefly, trajectories are aligned on the nAChR and positions of the lipid bead of interest are histogrammed on a grid with 1 Å spacing, resulting in a density measurement expressed in Å$^{-3}$. Upper and inner leaflets were defined by first identifying the membrane thickness based on average phosphate group positions, then dividing the membrane thickness into equal upper and lower portions. Figures of upper and lower densities are shown as summations along the z-axis of the density volume. Each leaflet thickness is approximately 16 Å. Cholesterol flip-flopping was detected using the tool LiPyPlilic[76].

For each simulation, we quantified the occupancy and interaction durations of each lipid with each residue in the nAChR TMD using the package PyLipID[77]. Occupancy is defined as the percentage of total simulation frames for which a lipid headgroup is within 5 Å of a coarse-grained "bead". To quantify interaction durations, a dual cut-off approach was used where an interaction was deemed to start when a lipid headgroup is within the lower cut-off distance (5 Å) of a residue and continues until the headgroup-residue distance exceeds the upper cut-off (8.5 Å). The cut-off distances were first established based on the interaction behavior of the PC headgroup with the nAChR in pure PC bilayers, based on the optimization protocol used in ref.[77] (Fig. S14). The lower cutoff was chosen based on the highest probability minimum distance from the lipid headgroup to any interacting residue, corresponding to specific lipid interactions, while the upper cutoff was chosen in order to account for the "rattling-in-cage" effect and to be able to sample long interactions. For consistency, the same cut-off values were then used to quantify the occupancy and interaction durations of each lipid in the multi-component lipid bilayers. See Supplementary Note 1 for a discussion of data convergence (Fig. S15).

The VMD program[78] was used for visualization of MD trajectories. Cavities in Cryo-EM structures were calculated using the KVFinder webtool[79]. The python package MDAnalysis[74,75] was used for protein-lipid distance measurements. Figures were created using VMD[78], ChimeraX[80] and the Matplotlib python package.

### Statistics and reproducibility
All occupancy and maximum duration averages and standard deviations are calculated from the three replicates performed for each CG simulation. Replicates were defined by re-generating the given nAChR/membrane

system with new lipid positions within the membrane. In atomistic simulations, replicates are defined by re-assigning initial atom velocities to the chosen backmapped frame at the beginning of equilibration. Lipid headgroup densities and patterns of high occupancy/duration are consistent between repeats, showing reproducibility.

## Reporting summary
Further information on research design is available in the Nature Portfolio Reporting Summary linked to this article.

## Data availability
All data supporting these findings are available within the paper and the Supplementary Information. Numerical source data for main text figures can be found in Supplementary Data 1. Initial coordinate and simulation input files and a coordinate file of the final output are deposited on Zenodo[82] (https://doi.org/10.5281/zenodo.10711467).

## Code availability
No custom code was used to generate or process data contained in this manuscript, the following open source software was used. GROMACS 2021.4 and NAMD 3.11 simulation software were used for CG and atomistic simulations respectively. Lipid occupancy and interaction duration was obtained using PyLipID v1.4.7. Lipid density and distance measurements were performed using MDAnalysis v2.0.0. Cholesterol flipflops were analyzed using LiPyPhilic v0.11.0. TMD cavities were measured using KVFinder v1.2.0.

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

## Acknowledgements

This work was supported by grants from CIHR (175223) and NSERC (113312) to J.E.B. F.D. acknowledges the State-Region Grand-Est Plan "Technological Innovations, Modeling and Personalized Medical Support" (IT2MP) and the European Regional Development Funds (ERDF) for generous support. This collaborative research was supported by a grant from the International Emerging Action program of the CNRS.

## Author contributions

A.A., F.D. and J.E.B. designed the research project. A.A. and R.Y.G. performed simulations and data analysis. A.A. and J.E.B. wrote the paper in consultation with F.D. A.A. and J.E.B. prepared figures with R.Y.G. preparing Fig. S13.

## Competing interests

The authors declare no competing interest.
