## [Peer review file · Communications Biology]

Reviewers' comments:

Reviewer #1 (Remarks to the Author):

-The authors should discuss if the MD simulations have been run only one time (1 replica) per system or if a multi-replica approach has been adopted in the study. If only 1 replica has been performed, the authors are invited to clarify their choice and explain in the article if the initial starting point could have influenced the results.

- Please explain how the lipid ratio composition has been determined. Have you considered any lipidomic analysis of the membrane under study?

-The authors should discuss in the manuscript if there is additional experimental information to corroborate their results.

-if possible the author should evaluate the change of CHOL distribution between the membrane leaflet (flipping) during CG simulations and if this behaviour can have implications on their findings

Reviewer #2 (Remarks to the Author):

Overall appraisal of the work

“Our data reconcile 40 years of biophysical data leading to a unified picture of lipid-nAChR interactions and set the stage for a detailed understanding of the mechanisms by which lipid binding influences function” (Abstract and lines 284-285); “...unprecedented insight into the nature of PA and Chol interactions” (lines 282-283). These and other grandiloquent statements contrast with the relatively few novel observations in the work of Ananchenko et al.

The study offers some insights into the structural aspects of lipid-nAChR interactions, but by no means provides the unified picture of this manifold topic that the authors claim. The cryo-EM study of Baenziger and Nury's group (Zarkadas et al., incomplete ref. 17) clashes in several aspects with the work of Hibbs and coworkers with regard to the location and nature of the lipid sites that are at the core of Ananchenko et al.'s work, which does not settle these disagreements.

Various conclusions drawn from the molecular dynamics simulations are very lightly elaborated, and do not resolve the above-mentioned discrepancies. The claimed “reconciliation of decades of biophysical data describing the nature of nAChR-lipid interactions” in the Abstract and reiterated in the main text, becomes an empty slogan.

Introduction

This section lists somewhat disconnected references to non-specific/specific effects of lipids on the nAChR, but lacks a clear connecting line of thought. For instance, the authors devote a long section to the possible effects of bulk lipids on the nAChR, qualifying these interactions as central (lines 37-40), yet the molecular dynamics do not address any aspects of such claimed central role.

Several authors have contributed to understanding how lipids modulate conformational transitions

of the nicotinic receptor. However, Ananchenko et al. give credit on this topic to work from their laboratory only. In general, the bibliography quoted appears to be biased towards the authors' production, and many references are incomplete.

Results

Section on Lipid organization at the nAChR-lipid interface (Lines 98-109).

Brings no new insights on the topography of this region beyond what has been elucidated in the recent cryo-EM studies of Baenziger and Hibbs' groups.

Section of PC exchanges rapidly... (lines 110-135)

Lines 110-117. Probably this paragraph constitutes the most interesting finding of the molecular simulation study, describing the rapidly exchanging phosphatidylcholine between "nAChR-bound sites" and the "bulk lipid", because the dynamics of this exchange have been experimentally studied in the past. The authors should provide quantitative information about this process and discuss the correlation between experimental work and molecular simulations in more depth.

Section on PA binding (lines 136-164)

This constitutes the second novel section of the work, illustrating the interaction of the functionally relevant phospholipid with the nAChR and its competition with PC.

Section on Cholesterol exhibits promiscuous long duration binding (lines 165-200)

Figure 2C: why aren't the nAChR TM chains shown as in A and B?

The section on state-dependent lipid interactions (lines 216-235) brings little if any new information. Lines 225-228. The "subtle" differences in cholesterol-nAChR interactions between unbound and agonist-bound nAChR have been analyzed in depth in cryo-EM studies, although no references to the previous work is provided, and there is no proper discussion of the MD findings in relation to the structural data.

Lines 229-230. The observation that cholesterol can "dive" into zones of the TMD close to the M2 helices lining the central ion channel, which the authors describe as a "new site", has been described in detail by Brannigan et al. (2008) and more recently by the group of Hibbs (Zhuang et al., 2022).

Lines 346-348: The same applies to the statement "The simulations highlight a propensity for the sterol ring to bind to the shallow grooves between TMD α -helices". The authors present these observations as original. The molecular dynamics studies of Brannigan et al. (2008) had already demonstrated this propensity. Ananchenko et al. do not quote this work.

Discussion

As previously stated, discussion and interpretation of the simulations in functional terms and their correlation to the recent cryo-EM data is insufficient.

Reviewer #3 (Remarks to the Author):

The manuscript by Ananchenko et al entitled State-Dependent Binding of Cholesterol and an Anionic Lipid to a Lipid-Sensitive Pentameric Ligand-Gated Ion Channel is well written and provides important

insight into lipid-protein interactions of the muscle type nAChR that have puzzled and engaged researchers for over 40 years. Importantly the molecular dynamic simulations are consistent with lipid binding identified in high resolution structures and the results explain biophysical data accumulated over the past 30-40 years. Most satisfying is the presence of cholesterol sites located in the grooves between adjacent transmembrane helices. This has long been what investigators have postulated and this study provides further evidence for this. I have no issues with manuscript or the findings. Just like the authors state the next important question to answer is what is the molecular mechanism by which the lipid interactions affect receptor function.

Reviewers' comments:

Reviewer #1 (Remarks to the Author):

-The authors should discuss if the MD simulations have been run only one time (1 replica) per system or if a multi-replica approach has been adopted in the study. If only 1 replica has been performed, the authors are invited to clarify their choice and explain in the article if the initial starting point could have influenced the results.

We used a multi-replica approach for our MD simulations with three repeats of 30 μ s for each of the coarse-grained simulations and two repeats of 250 ns for each of the all-atom simulations. We have added statements to the Results section of the revised manuscript to emphasize the multi-replica approach:

Lines 83-86: *“As a first step towards understanding the mechanisms by which neutral and anionic lipids modulate Torpedo nAChR function, we performed 3 x 30 μ s long CG-MD simulations with both apo and nicotine-bound nAChR structures imbedded in pure PC, 3:2 PC:PA, 3:2 PC:Chol, and 3:1:1 PC:PA:Chol membranes (all molar ratios).”*

Lines 250-252: *“To explore the nature of the physical interactions that drive lipid binding, we back mapped select frames from the apo and nicotine-bound CG-MD simulations and performed 2 x 250 ns long all atom simulations.”*

- Please explain how the lipid ratio composition has been determined. Have you considered any lipidomic analysis of the membrane under study?

We apologize for not providing a clear enough rationale for the choice of lipids used in our simulations. We have revised the text to state:

Lines 32-38: *“Decades of biophysical and biochemical studies have established that the activity of the prototypic pLGIC, the muscle-type nicotinic acetylcholine receptor (nAChR) from Torpedo, is sensitive to a variety of lipids, including neutral lipids, such as cholesterol (Chol), and anionic lipids, such as phosphatidic acid (PA)¹⁰⁻¹². Although the nAChR exhibits only weak structural specificity for both lipid types^{13,14}, PA is particularly effective at stabilizing an agonist-responsive conformation¹⁵⁻¹⁷. Also, a 3:1:1 molar ratio of phosphatidylcholine (PC), PA, and Chol is as effective as native membranes in stabilizing an agonist-responsive nAChR^{12,18}.”*

Lines 76-80: *“We focus on simple bilayers containing PC, PA and Chol whose effects on nAChR function have been characterized in vitro, with the goal of elucidating how PA and Chol, two lipids that support conformational transitions, both individually and synergistically compete for binding to the nAChR with PC, a lipid that does not support conformational transitions.”*

Lines 86-98: *“These lipid mixtures were chosen for three reasons. First, the agonist-induced response of the nAChR in each of the membranes has been characterized in vitro (Fig. 1). The nAChR in PC-only membranes is stabilized in an uncoupled conformation that normally does not undergo agonist-induced conformational transitions¹⁸. The 3:2 molar ratios of PC:PA and PC:Chol are the optimal ratios of both PA and Chol in a PC membrane for stabilizing an agonist-responsive nAChR, although mixtures of PC and PA are more effective in this regard (Fig. 1)¹³. The 3:1:1 PC:PA:Chol membrane is as effective as native membranes at stabilizing an agonist-responsive nAChR and thus serves as a surrogate for a native*

membrane environment (Fig. 1c). Second, simulations of the nAChR in these four simple membranes allow us to directly assess how PA and Chol, lipids that particularly effective at supporting an agonist-induced response in reconstituted membranes, both individually and synergistically compete for binding to the nAChR with PC, a lipid that does not support an agonist-induced response. Third, the relatively high levels of PA and Chol enhance the sampling of lipid binding to provide unprecedented insight into the dynamic nature of PA and Chol interactions.”

Lines 291-297: *“Here, we have taken a unique multiscale MD simulation approach to study lipid binding to the nAChR in that we have focused on simple, defined membranes containing levels of PC, PA and/or Chol that are known to stabilize the nAChR in reconstituted membranes in different conformational states. These simulations allow us to correlate lipid binding patterns directly with documented effects of lipids on the function of the nAChR in the reconstituted liposomes. Specifically, we set out to understand how lipids, PA and Chol, that support an agonist-induced response compete for binding to the nAChR with a lipid, PC, that does not. The relatively high levels of PA and Chol also enhance the sampling of lipid binding.”*

The goal of this study was mechanistic, to address how PA and Chol influence function in reconstituted liposomes. We are performing a lipidomics analysis of the *Torpedo* membranes and are in the process of extending the MD simulations to biological membranes.

-The authors should discuss in the manuscript if there is additional experimental information to corroborate their results.

As noted, we have revised the Introduction to contain a more robust discussion of the experimental data underlying our understanding of lipid-nAChR interactions. We have also expanded the discussion of our findings with particular attention to comparing the lipid binding observed in the MD simulations to the lipid binding observed in recent cryo-EM structures of the nAChR (see Figures S3 and S12).

-if possible the author should evaluate the change of CHOL distribution between the membrane leaflet (flipping) during CG simulations and if this behaviour can have implications on their findings

We have examined the flip flop of Chol in the membrane and did not observe any change in the distribution of Chol between the two leaflets over the course of the simulations, as now shown in Fig. S11.

Reviewer #2 (Remarks to the Author):

Overall appraisal of the work

“Our data reconcile 40 years of biophysical data leading to a unified picture of lipid-nAChR interactions and set the stage for a detailed understanding of the mechanisms by which lipid binding influences function” (Abstract and lines 284-285); “...unprecedented insight into the nature of PA and Chol interactions” (lines 282-283). These and other grandiloquent statements contrast with the relatively few novel observations in the work of Ananchenko et al.

We agree and have changed the *abstract to state:*

Lines 25-27: *“Our data provide insight into the dynamic nature of lipid-nAChR interactions and set the stage for a more detailed understanding of the mechanisms by which lipids facilitate nAChR function at the neuromuscular junction.”*

We have revised the final sentence of the Introduction to state:

Lines 80-81: *“Our simulations reveal state-dependent interactions at subunit specific sites thus providing new insight into the mechanisms by which lipids influence nAChR function.”*

We have revised the Discussion to state:

Lines 300-302: *“The resulting data not only shed light on the mechanisms by which lipids modulate function but reconcile observations obtained from numerous biophysical studies probing the nature of nAChR-lipid interactions.”*

The study offers some insights into the structural aspects of lipid-nAChR interactions, but by no means provides the unified picture of this manifold topic that the authors claim.

As noted, we have revised the text to focus the discussion on the important findings of this work.

The cryo-EM study of Baenziger and Nury’s group (Zarkadas et al., incomplete ref. 17) clashes in several aspects with the work of Hibbs and coworkers with regard to the location and nature of the lipid sites that are at the core of Ananchenko et al.’s work, which does not settle these disagreements.

We disagree with this statement. The structures from our lab and the Hibbs lab are complementary because each provides a snapshot of lipid binding under a specific experimental condition, with precise conditions varying from one structure to another. All things being equal, the number of modeled lipids depends on the resolution of the density maps, which varies from structure to structure. The nature and positions of the modeled lipids also depend on the methods of sample preparation - our structures were solved using the nAChR purified in the *presence* of added exogenous soybean azolectin to facilitate lipid exchange and thus contain no modeled endogenous lipids while the Hibbs’ structures were solved using the nAChR purified in the *absence* of exogenous lipids and thus contain endogenous Chol. Hibbs also added exogenous Cho to one sample (PDB 7SMQ; apo+Chol) leading to the observation of additional Chol binding sites. The effect of sample preparation conditions on the modeled lipids can be seen clearly comparing the two Hibbs structures prepared with and without added exogenous Chol (Fig. S3). The former structure models 11 and 8 Chol and PC lipids, respectively, while the latter models only 3 and 8 Chol and PC lipids – differences clearly attributed to the method of sample preparation. Furthermore, the type of scaffolding protein used to prepare the nAChR nanodiscs used for imaging likely affects the number and positions of the bound lipids, as discussed in Ananchenko et al., *Biomolecules* 2022 Jun 10;12(6):814. doi: 10.3390/biom12060814.

There are no “disagreements” to “settle” between the two structures. In fact, the variations in lipid binding from one structure to another simply show that lipids bound to the nAChR can be exchanged for other lipids leading to different lipid binding poses in different structures. This finding is entirely consistent with our MD simulations, which characterize the dynamic nature of lipid binding to the apo and agonist-bound conformations. We show that bound lipids undergo rapid exchange with those in the bulk membrane. We have generated a new Figure S3, with an accompanying discussion, to better integrate the data obtained from the structural and simulation techniques.

conclusions drawn from the molecular dynamics simulations are very lightly elaborated, and do not resolve the above-mentioned discrepancies. The claimed “reconciliation of decades of biophysical data describing the nature of nAChR-lipid interactions” in the Abstract and reiterated in the main text, becomes an empty slogan.

As noted, we have revised the text extensively. We added Figure S3 to show the complementarity between the structural and MD simulation data.

Introduction

This section lists somewhat disconnected references to non-specific/specific effects of lipids on the nAChR, but lacks a clear connecting line of thought. For instance, the authors devote a long section to the possible effects of bulk lipids on the nAChR, qualifying these interactions as central (lines 37-40), yet the molecular dynamics do not address any aspects of such claimed central role.

We have revised the Introduction to focus solely on lipid binding to the nAChR. We have added a historical context with regards to previous MD simulations, highlighting the necessity for MD simulations using the higher resolution correct apo and agonist bound structures.

Several authors have contributed to understanding how lipids modulate conformational transitions of the nicotinic receptor. However, Ananchenko et al. give credit on this topic to work from their laboratory only. In general, the bibliography quoted appears to be biased towards the authors' production, and many references are incomplete.

We focused on functional studies that directly examine the competing and synergistic roles of PC, PA, and Chol as these are the lipids chosen for this MD simulation study. We have added citations along with a brief historical summary of previous MD simulations performed on the *Torpedo* nAChR.

Results

Section on Lipid organization at the nAChR-lipid interface (Lines 98-109).

Brings no new insights on the topography of this region beyond what has been elucidated in the recent cryo-EM studies of Baenziger and Hibbs' groups.

We do not understand the concern. The referred to paragraph is only an introduction to the simulation data that lays the groundwork for the more detailed discussion of lipid binding presented subsequently in the Results.

We re-emphasize that the simulations provide insight into the dynamic nature of lipid binding as opposed to the cryo-EM data, which present snapshots of bound lipids under specific experimental conditions after *rapid freeze-plunging to cryo-temperatures*. Furthermore, we compare how the binding of two lipids, PA and Chol, that support nAChR function in reconstituted liposomes, compete individually and synergistically with PC, a lipid that does not support nAChR function. These questions are not addressed in the cryo-EM structures. These questions are critical to understanding how lipids influence nAChR function.

Section of PC exchanges rapidly... (lines 110-135)

Lines 110-117. Probably this paragraph constitutes the most interesting finding of the molecular simulation study, describing the rapidly exchanging phosphatidylcholine between "nAChR-bound sites" and the "bulk lipid", because the dynamics of this exchange have been experimentally studied in the past. The authors should provide quantitative information about this process and discuss the correlation between experimental work and molecular simulations in more depth.

To provide quantitative insight into the dynamics of lipid binding, we present interaction duration histograms in both Figs. 4 and 5. To the best of our knowledge, no MD simulation paper has ever presented such interaction duration histograms in order to dive deeply into the simulation data. We have also added the following statement:

Lines 307-310: “In addition, the electron spin resonance experiments estimate rates of exchange between the bulk and immobilized lipids between $10^5 s^{-1}$ and $5 \times 10^7 s^{-1}$ ^{43,45}, which are consistent with the ns to μs lipid residence times detected in the MD simulations (Figs. 4 and 5).”

Section on PA binding (lines 136-164)

This constitutes the second novel section of the work, illustrating the interaction of the functionally relevant phospholipid with the nAChR and its competition with PC.

We thank the reviewer for this comment. This study provides the first insight into how PA, a lipid with unique abilities to stabilize the nAChR in an agonist-responsive state, interacts with the nAChR and competes with PC for binding. Although not noted by the reviewer, we also compared the binding of mono-anionic versus di-anionic PA to assess how the ionization state of PA influences nAChR binding – as the ionic state of PA has been suggested to play an important role in how it influences nAChR function.

Section on Cholesterol exhibits promiscuous long duration binding (lines 165-200)

Figure 2C: why aren't the nAChR TM chains shown as in A and B?

We originally did not include the TM chains in the 2D headgroup density plots for membranes containing Chol as the TMD helices obscured some of the headgroup density located near the middle of the bilayer. To resolve this issue, we now superimpose TM chains that are slightly transparent so that all Chol headgroup densities can be seen in the figure.

The section on state-dependent lipid interactions (lines 216-235) brings little if any new information. Lines 225-228. The “subtle” differences in cholesterol-nAChR interactions between unbound and agonist-bound nAChR have been analyzed in depth in cryo-EM studies, although no references to the previous work is provided, and there is no proper discussion of the MD findings in relation to the structural data.

We disagree. This study provides the first insight into the *dynamic nature* of lipid binding to apo versus agonist-bound states. As noted, the lipid binding patterns observed in cryo-EM structures are only snapshots of lipid binding under specific experimental conditions after rapid vitrification. For this reason, it is difficult to differentiate between state-dependent lipid binding poses versus simple variations in binding poses that typically occur from one structure to another. We illustrate these points in the new Figure S3.

On the other hand, both our and the Hibbs structures reveal a tilting of M4 in the agonist-bound state that could affect lipid binding to the agonist-bound state. The Hibbs group did investigate the potential role of residues at the M4–M1/M3 interface in desensitization along with the binding of curare to this interface. We address the state dependent M4 tilting in the revised manuscript.

Lines 247-250: “Finally, although previous structures have revealed a tilting of the M4 α -helices in the agonist-bound state, which could lead to enhanced lipid binding at the M4-M1/M3 interface, we do not detect major changes in either the lipid occupancy or interaction durations at this interface in the nicotine bound state”

Lines 229-230. The observation that cholesterol can “dive” into zones of the TMD close to the M2 helices lining the central ion channel, which the authors describe as a “new site”, has been described in detail by Brannigan et al. (2008) and more recently by the group of Hibbs (Zhuang et al., 2022).

The reviewer's interpretation of the Brannigan et al. study is not accurate. Brannigan et al. manually and with docking programs *placed* Chol into gaps between the TMD α -helices in the 2BG9 *Torpedo* structure and then performed all-atom simulations to examine the effects of the inserted Chol on the stability of the structure. They did not observe Chol diving into the membrane with its headgroup projecting towards M2. In fact, they did not probe the *dynamics* of Chol binding to the TMD. Furthermore, while the study of Brannigan et al. was rigorous and state-of-the art at that time, it unavoidably used the 4 Å resolution 2BG9 *Torpedo* structure, which exhibits errors including a scaling error that may have over emphasized the size of the gaps between the TMD α -helices. To address this concern, we now include a new supplemental figure comparing the size of the gaps that were used by Brannigan to dock Chol in the 2BG9 structure to the gaps in the new higher resolution structures (Fig. S12). The comparison shows that Chol cannot fit into the gaps between TMD helices as originally suggested by Brannigan et al.

The study of Zhuang et al. reported that Chol dives into the membrane to bind to a known positive allosteric modulator site on the TMD of the homomeric $\alpha 7$ nAChR – a binding pose similar to that observed by us for Chol binding to the *Torpedo* nAChR in the nicotine-bound state. The Zhuang et al. paper does not present any data pertaining to lipid binding to the muscle type *Torpedo* nAChR. Our observation of a similar binding pose in the *Torpedo* nAChR is thus entirely novel. Most importantly, the complementary data obtained here and by Zhang et al. lend support to the potential importance of this site/these interactions more broadly in nAChR function. We cited the work of Zhuang et al. in the first submitted version of the manuscript, as we do in the revised version (reference 39).

Lines 346-348: The same applies to the statement “The simulations highlight a propensity for the sterol ring to bind to the shallow grooves between TMD α -helices”. The authors present these observations as original. The molecular dynamics studies of Brannigan et al. (2008) had already demonstrated this propensity. Ananchenko et al. do not quote this work.

This statement does not accurately reflect the findings of Brannigan et al. Brannigan et al. either manually or using docking programs placed Chol into the grooves between TMD α -helices and then performed all-atom simulations. While the docking programs do highlight preferred interacting poses in the grooves between the TMD α -helices, Brannigan et al. did not independently assess the dynamic nature of Chol binding to the nAChR surface and thus did not compare the propensity for Chol to bind to grooves between TMD α -helices versus other regions on the TMD surface. Furthermore, an important observation from our simulations is that Chol binds deeper in the inter α -helix grooves in a manner that does not impact on PC binding (Fig. S9). Regardless, as noted above, although the study of Brannigan et al. was pioneering at that time, it unavoidably based its interpretations on simulations using the 2BG9 structure, which now limits the conclusions that can be derived from the simulations.

Discussion

As previously stated, discussion and interpretation of the simulations in functional terms and their correlation to the recent cry-EM data is insufficient.

We have revised the manuscript extensively to interpret the data in functional terms and to correlate the simulations with the structural data.

Reviewer #3 (Remarks to the Author):

The manuscript by Ananchenko et al entitled State-Dependent Binding of Cholesterol and an Anionic Lipid to a Lipid-Sensitive Pentameric Ligand-Gated Ion Channel is well written and provides important insight into lipid-protein interactions of the muscle type nAChR that have puzzled and engaged researchers for over 40 years. Importantly the molecular dynamic simulations are consistent with lipid

binding identified in high resolution structures and the results explain biophysical data accumulated over the past 30-40 years. Most satisfying is the presence of cholesterol sites located in the grooves between adjacent transmembrane helices. This has long been what investigators have postulated and this study provides further evidence for this. I have no issues with manuscript or the findings. Just like the authors state the next important question to answer is what is the molecular mechanism by which the lipid interactions affect receptor function.

We thank the reviewer for these comments.

Reviewers' comments:

Reviewer #3 (Remarks to the Author):

Accept revised manuscript - intriguing work!

Reviewer #4 (Remarks to the Author):

This work from the Baenziger group surveys lipid interactions with the Torpedo nicotinic receptor. The team uses long timescale MD simulations starting from recent high resolution cryo-EM structures of the receptors. I find the study to be clearly written, fairly referenced, and with clear figures. I enjoyed reading it. I do not think it adds much to our understanding of how lipids regulate the activity of the Torpedo nicotinic receptor, as it is purely theoretical. No experimental validation is present in the manuscript. The gold standard would be to reconstitute the channel in lipids and do an electrophysiology experiment to test whether, indeed, channel activity is absent in PC, rescued by PA, and potentiated by cholesterol, then make mutants to knock out a key hypothesized site, and repeat. That the channel is robustly functional in asolectin and soy polar lipids, which lack cholesterol and probably most plant sterols, suggests that the role of cholesterol may be overstated in the literature. I was brought into the review process after the first round of review. I was asked to comment on whether I think the authors satisfyingly addressed the first reviewers' comments, and whether I had any other questions/comments/concerns. My sense is thus that the journal and editor are happy with the scope and advance, which is fine. I will try to keep my comments limited, accordingly.

First, I carefully read the feedback from reviewers in the first round, and I think the authors did a good job in responding. I have no concerns with their responses.

Specific comments/requests:

1. For clarity, please state explicitly that the study is on the Torpedo nicotinic receptor in the abstract, and maybe also the title, but definitely the abstract.
2. Line 21 in the abstract uses the word "reconstituted," which made me think we would see some electrophysiology on reconstituted receptors, which would be fantastic- really, the best way to test hypotheses from this theoretical study. I understand that these are beyond the scope of the current study. However, please remove this terminology that implies a wet bench experiment to measure channel function.
3. Line 24, typo, missing "by" before "a key."
4. You might replace the termed "solved" for cryo-EM structures with "determined." Solved refers (usually) to solving the electron density equation in x-ray crystallographic experiments. No such thing occurs in single particle cryo-EM.
5. Line 67, and maybe this is just my bias, but I do not think MD simulations can "answer detailed questions." I think they can provide excellent relative thermodynamic information to help guide understanding and designing of experiments to test hypotheses. The detailed question about why PA is ~required for Torpedo nAChR activity is still unanswered here, though the MD simulations suggest where it might bind and for how long. You might just tone down this statement, if you agree.
6. Lines 300-301, as for line 67, I think this wording overstates the advance. How does PA binding to a specific site enable channel function? I do not know, from reading this study.
7. Line 394. The nicotine bound cryo-EM structure is stated as being in an "activated" state. Please change this term. As the authors are likely aware, there is a big mess in the channel structure field in trying to accurately annotate the functional state of experimental structures. The problem is particularly severe for cation-selective Cys-loop receptors, which have polar residues where the

desensitization 'gate' is expected to be. Anion-selective Cys-loop receptors have hydrophobic residues there, offering an easier to understand barrier to ion permeation. Electrophysiology sheds light on how big pores should be when "open." MD simulations can suggest whether a pore is partially/fully hydrated and how many ions might go through per second, though absolute conductance measurements are unreliable by MD. Electrophysiological measurements of channel kinetics tell us that barring a major sample problem, in the presence of nicotine, the structure should be, at equilibrium, desensitized. To be open, the pore should be larger (by Hille's measurements). The original paper from Nury and Baenziger groups called the nicotine-bound structure "desensitized-like," because the channel should (from electrophysiology) be closed, it looks partially closed, it does not look like published activated state structures, but in MD simulations it is partially hydrated and rarely occupied by ions. Calling it "activated" is very confusing without dealing with all of these important caveats, which the field as a whole does need to come up with a way to deal with. I respectfully request that you change the term from activated to something else (perhaps what it was originally called, desensitized-like).

8. Line 396, a question about the MD setup. Why was nicotine removed- was this necessary? Removing nicotine, then running a long time scale simulation, may result in a conformational shift toward the resting state- it should. If one wants to study an agonist-bound conformation, why not leave the agonist in? Perhaps I have misunderstood the logic of this MD experiment.

Fig. S10, typo in title.

Reviewers' comments (response in green):

Reviewer #3 (Remarks to the Author):

Accept revised manuscript - intriguing work!

We thank the reviewer for his/her assessment.

Reviewer #4 (Remarks to the Author):

This work from the Baenziger group surveys lipid interactions with the Torpedo nicotinic receptor. The team uses long timescale MD simulations starting from recent high resolution cryo-EM structures of the receptors. I find the study to be clearly written, fairly referenced, and with clear figures. I enjoyed reading it. I do not think it adds much to our understanding of how lipids regulate the activity of the Torpedo nicotinic receptor, as it is purely theoretical. No experimental validation is present in the manuscript. The gold standard would be to reconstitute the channel in lipids and do an electrophysiology experiment to test whether, indeed, channel activity is absent in PC, rescued by PA, and potentiated by cholesterol, then make mutants to knock out a key hypothesized site, and repeat. That the channel is robustly functional in asolectin and soy polar lipids, which lack cholesterol and probably most plant sterols, suggests that the role of cholesterol may be overstated in the literature. I was brought into the review process after the first round of review. I was asked to comment on whether I think the authors satisfyingly addressed the first reviewers' comments, and whether I had any other questions/comments/concerns. My sense is thus that the journal and editor are happy with the scope and advance, which is fine. I will try to keep my comments limited, accordingly.

First, I carefully read the feedback from reviewers in the first round, and I think the authors did a good job in responding. I have no concerns with their responses.

Thank you!

Specific comments/requests:

1. For clarity, please state explicitly that the study is on the Torpedo nicotinic receptor in the abstract, and maybe also the title, but definitely the abstract.

We agree with this suggestion. We have changed the title of the manuscript to:

"State-dependent binding of cholesterol and an anionic lipid to the muscle-type Torpedo nicotinic acetylcholine receptor"

We have also revised the first sentence of the Abstract to read:

"The ability of the Torpedo nicotinic acetylcholine receptor (nAChR) to undergo agonist-induced conformational transitions..."

2. Line 21 in the abstract uses the word “reconstituted,” which made me think we would see some electrophysiology on reconstituted receptors, which would be fantastic- really, the best way to test hypotheses from this theoretical study. I understand that these are beyond the scope of the current study. However, please remove this terminology that implies a wet bench experiment to measure channel function.

We removed the statements referring to “reconstituted” membranes from the abstract to focus the discussion on the computational findings. We note, however, that we did/do include wet-lab functional measurements from the nAChR reconstituted into different membranes in Figure 1. These functional measurements provided the foundation for the presented computational work.

3. Line 24, typo, missing “by” before “a key.”

We added the missing “by”.

4. You might replace the termed “solved” for cryo-EM structures with “determined.” Solved refers (usually) to solving the electron density equation in x-ray crystallographic experiments. No such thing occurs in single particle cryo-EM.

True. We replaced the word “solved” (line 43) with “determined”.

5. Line 67, and maybe this is just my bias, but I do not think MD simulations can “answer detailed questions.” I think they can provide excellent relative thermodynamic information to help guide understanding and designing of experiments to test hypotheses. The detailed question about why PA is ~required for Torpedo nAChR activity is still unanswered here, though the MD simulations suggest where it might bind and for how long. You might just tone down this statement, if you agree.

We agree. We changed the sentence (line 67) from “...answer detailed questions...” to “explore”.

6. Lines 300-301, as for line 67, I think this wording overstates the advance. How does PA binding to a specific site enable channel function? I do not know, from reading this study.

We agree. We have revised the sentence (lines 300-302) from:

“The resulting data not only shed light on the mechanisms by which lipids modulate function but reconcile observations obtained from numerous biophysical studies...”

to now read:

“The resulting data not only set the stage for functional measurements, which should provide detailed insight into the mechanisms by which lipids facilitate nAChR function at the neuromuscular junction, but also reconcile several observations obtained from previous biophysical studies...”

We have revised other statements in the manuscript to highlight that our data is hypothesis generating, not hypothesis confirming. For example, on lines 381-382 we summarize the Discussion of lipid sites by stating:

“The mechanistic importance of these and other observed lipid binding sites must be verified by functional measurements.”

7. Line 394. The nicotine bound cryo-EM structure is stated as being in an “activated” state. Please change this term. As the authors are likely aware, there is a big mess in the channel structure field in trying to accurately annotate the functional state of experimental structures. The problem is particularly severe for cation-selective Cys-loop receptors, which have polar residues where the desensitization ‘gate’ is expected to be. Anion-selective Cys-loop receptors have hydrophobic residues there, offering an easier to understand barrier to ion permeation. Electrophysiology sheds light on how big pores should be when “open.” MD simulations can suggest whether a pore is partially/fully hydrated and how many ions might go through per second, though absolute conductance measurements are unreliable by MD. Electrophysiological measurements of channel kinetics tell us that barring a major sample problem, in the presence of nicotine, the structure should be, at equilibrium, desensitized. To be open, the pore should be larger (by Hille’s measurements). The original paper from Nury and Baenziger groups called the nicotine-bound structure “desensitized-like,” because the channel should (from electrophysiology) be closed, it looks partially closed, it does not look like published activated state structures, but in MD simulations it is partially hydrated and rarely occupied by ions. Calling it “activated” is very confusing without dealing with all of these important caveats, which the field as a whole does need to come up with a way to deal with. I respectfully request that you change the term from activated to something else (perhaps what it was originally called, desensitized-like).

We agree. In the text (line 226-227) of the Results where we discuss state-dependent binding, we refer to the nicotine bound state as follows:

“the nicotine-bound structure of the nAChR, which has a capped loop C around the agonist and an open Leu9’/Val13’ gate”

In line 394 of the Experimental procedures, we now refer to the nicotine bound state as:

“...nicotine-bound (desensitized-like state with a capped loop C around the agonist and an open Leu9’/Val13’ gate, see Zarkadas et al.²¹, PDB: 7QL5)...”

Note that we refer the reader to Zarkadas et al. to alert the reader to the fact that the conformation of the nicotine bound state is not definitive.

8. Line 396, a question about the MD setup. Why was nicotine removed- was this necessary? Removing nicotine, then running a long time scale simulation, may result in a conformational shift toward the resting state- it should. If one wants to study an agonist-bound conformation, why not leave the agonist in? Perhaps I have misunderstood the logic of this MD experiment.

Nicotine was not included in the MD simulations because we do not have force field parameters for nicotine, so it is not a trivial matter to simulate nicotine in the binding site. Fortunately, the absence of nicotine from the simulations has no impact on our findings for two reasons. First, in the CG-MD simulations, an elastic network is applied to the nAChR with the purpose of restraining the nAChR’s tertiary and quaternary structure to the nicotine-bound state. This approach is standard practice for CG-MD simulations in part because coarse graining eliminates backbone hydrogen bonding, etc. The elastic network ensures that the desensitized-like state is preserved throughout the CG-MD simulation trajectories, with no need for a bound agonist to be present. Second, the atomistic simulations, which were performed on a structure back-mapped from the CG-MD simulations, are too short to observe a biologically relevant conformational change in the protein. For both reasons, bound nicotine is not required to sample lipid interactions with the desensitized-like state.

Fig. S10, typo in title.

This has been fixed.

REVIEWERS' COMMENTS:

Reviewer #4 (Remarks to the Author):

I appreciate the fast and thoughtful responses/revisions from the authors. I have no further critiques.